# Topological descriptors of the parameter region of multistationarity: Deciding upon connectivity

**Máté László Telek, Elisenda Feliu** [ID] *

Department of Mathematical Sciences, University of Copenhagen, Copenhagen, Denmark

\* efeliu@math.ku.dk

**Data Availability Statement:** The source code for the computations in the manuscript is available as supplementary material.

**Funding:** This study has been funded by the Independent Research Fund of Denmark, under the

## Abstract

Switch-like responses arising from bistability have been linked to cell signaling processes and memory. Revealing the shape and properties of the set of parameters that lead to bistability is necessary to understand the underlying biological mechanisms, but is a complex mathematical problem. We present an efficient approach to address a basic topological property of the parameter region of multistationary, namely whether it is connected. The connectivity of this region can be interpreted in terms of the biological mechanisms underlying bistability and the switch-like patterns that the system can create. We provide an algorithm to assert that the parameter region of multistationarity is connected, targeting reaction networks with mass-action kinetics. We show that this is the case for numerous relevant cell signaling motifs, previously described to exhibit bistability. The method relies on linear programming and bypasses the expensive computational cost of direct and generic approaches to study parametric polynomial systems. This characteristic makes it suitable for mass-screening of reaction networks. Although the algorithm can only be used to certify connectivity, we illustrate that the ideas behind the algorithm can be adapted on a case-by-case basis to also decide that the region is not connected. In particular, we show that for a motif displaying a phosphorylation cycle with allosteric enzyme regulation, the region of multistationarity has two distinct connected components, corresponding to two different, but symmetric, biological mechanisms.

## Author summary

This work addresses the challenging problem of studying the set of parameters for which a system of ordinary differential equations has more than one steady state, a property termed multistationarity. In particular, we are interested in systems arising from the study of biochemical networks. The shape of the multistationarity region is linked to different types of switches that the network can display. We provide an algorithm to decide whether this set is path connected, meaning that any two points in the set are joined by a path completely contained in the set. We illustrate the algorithm with numerous relevant networks, for which we can conclude that the parameter region is path connected.

grant agreement with reference 0135-00090B to EF. The funder had no role in study design, data collection and analysis, decision to publish, or preparation of the manuscript.

**Competing interests:** The authors have declared that no competing interests exist.

This is a *PLOS Computational Biology* Methods paper.

## Introduction

Bistable switches are frequently observed and studied in living systems, and have been linked to cellular decision making and memory processes [1, 2]. These switches arise in different forms; one common form in parametric systems is that of *hysteresis* [3], that is, the system is monostable for small or large values of a parameter, and has two or more stable steady states for intermediate values. When the parameter changes slowly enough to allow the system to remain approximately at steady state, the resulting steady states depend on whether the parameter is increased or decreased. This is illustrated in Fig 1(a), which displays a hypothetical system with three steady states. When the parameter increases from low to high, the steady state goes through a bifurcation at a critical parameter value $\tau_{\max}$, after which the system discontinuously settles to another region of the output space. If the parameter value is decreased again, the system remains at the high steady state value, that is, it does not return to the low steady state value immediately. This will first happen if the parameter is decreased beyond another critical value $\tau_{\min} < \tau_{\max}$. This behavior confers the switch with *robustness*: after a change of level of steady state value takes place, small fluctuations in the parameter will not reverse the change. The larger the interval $[\tau_{\min}, \tau_{\max}]$ where the system has several steady states, the higher the degree of robustness of the change. More complicate switches can arise if, for example, the system has more than one steady state (is *multistationary*) in two intervals of the parameter, as illustrated in Fig 1(b) and 1(c). Irreversible switches are obtained if the relevant interval is of the form $(0, \tau_{\max}]$. Fig 1(a), 1(b) and 1(c) are qualitatively different in one important aspect: the parameter region where the system has multistationarity is connected in Fig 1(a) and disconnected (has two disjoint pieces) in Fig 1(b) and 1(c).

This phenomenon appears also in higher dimensions, where instead of a curve of steady states, there is a steady state manifold with "bends", and instead of having one parameter being

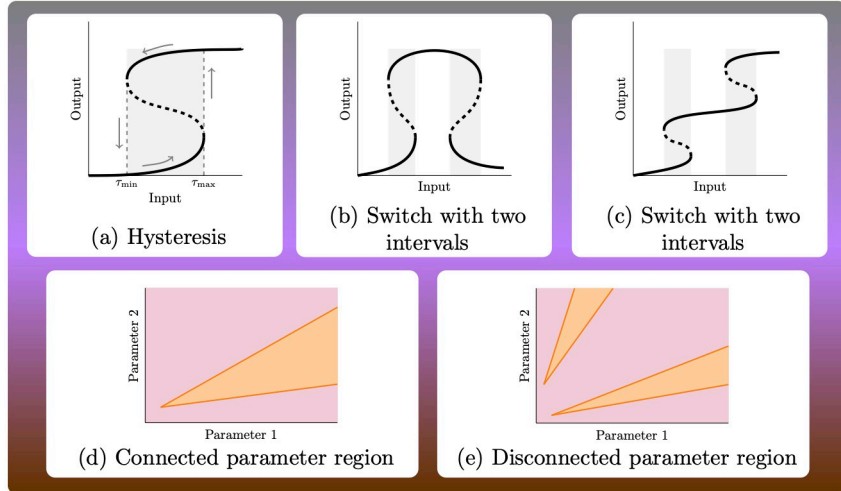

**Fig 1.** (a-c) Input-output curves for hypothetical systems. Input is thought to be a parameter of the system that is varied, and the output is the concentration of a species at steady state. Dashed lines correspond to unstable steady states, and solid lines to stable steady states. (a) displays a simple hysteresis switch; (b-c) show input-output curves for systems where bistability arises in two disjoint intervals of input. (d-e) For a system with two parameters, the system has more than one positive steady state in the orange regions, and one in the purple regions. In panel (d) the multistationarity region is connected, while in (e) it has two connected components.

varied, a vector of parameters is changed along a curve. The shape of the *parameter region of multistationarity* (or *multistationarity region* for short), and specifically the number of path-connected components, modulates the type of switches that can arise. If the multistationarity region is path connected, as in Fig 1(d), then any two parameter values in the region can be joined by a continuous path completely included in the region, typically giving rise to simple hysteresis switches (if the system has three steady states for parameters in the region). If, on the contrary, the region has two path-connected components, as in Fig 1(e), then any path joining two parameter points in different regions, necessarily goes through parameter points where the system does not have several steady states, allowing for complex switches to arise.

Mathematically, understanding the connectivity of a region is a basic topological property of a set, and the number of connected components is called the 0th Betti number of the set. Higher order Betti numbers describe the shape of the set in more detail, for instance, the first Betti number is the number of "holes" of the set. Tools from topological data analysis can infer the Betti numbers of a set from sample data points. By generating points in the multistationarity region, properties of the shape of the region have been explored for a specific dual phosphorylation system (Fig 2(h)) in [4], where it has also been suggested that lack of connectivity may indicate that different biological mechanisms underlie multistationarity.

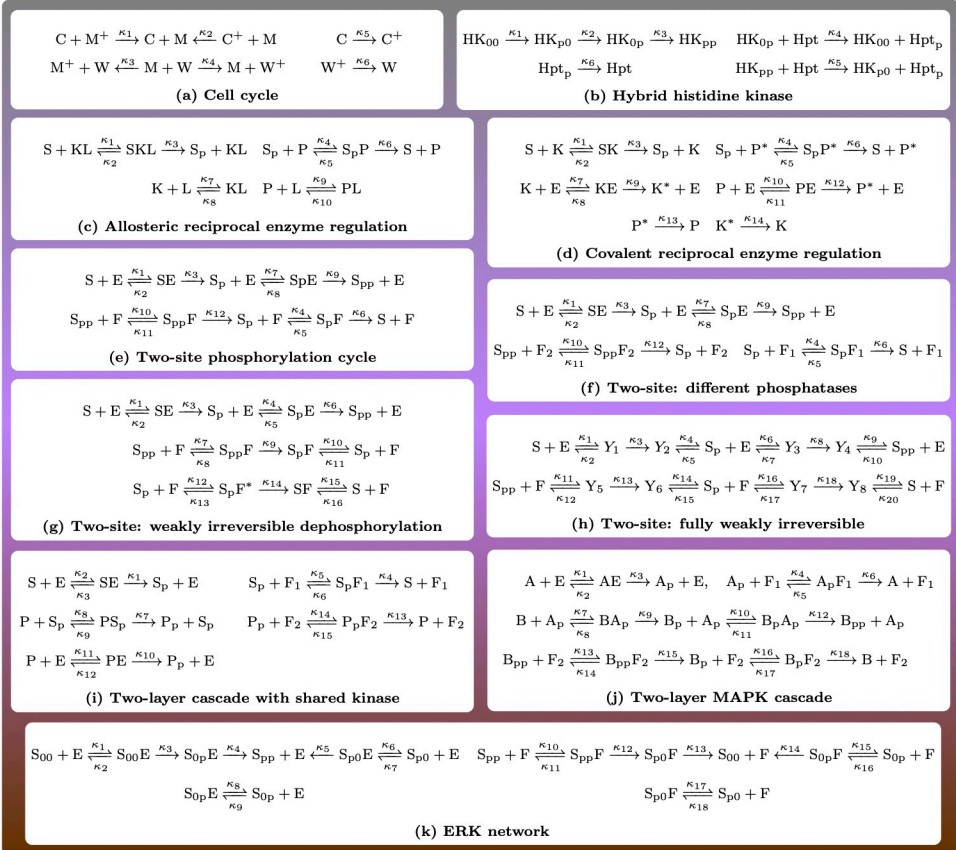

**Fig 2. Reaction networks arising in cell signaling.** The subindex 'p' indicates a phosphorylated site. When writing 'p' and 'pp' it is assumed the substrate has two phosphorylation sites, and phosphorylation/dephosphorylation is ordered. When writing '0p' for example, it means the substrate also has two sites numbered 1 and 2, and the second one is phosphorylated. All networks are known to be multistationary. For all networks but (c) and (h), the multistationarity region is path connected. For network (c), the multistationarity region has two path-connected components, while for network (h) our approach is inconclusive.

In this work, we address connectivity of the multistationarity region for polynomial systems describing the steady states of biochemical reaction networks. We achieve this by using exact symbolic tools and theoretical results relating the multistationarity region with the region where a polynomial attains negative values. Specifically, we work in the framework of chemical reaction network theory [5, 6], where extensive work has been done to decide whether a network exhibits multistationarity, i.e. whether the multistationarity region is non-empty [7]. More recent work focuses on understanding and finding the multistationarity region, but here progress is scarce and often restricted to special systems, e.g. [8–14].

Finding and studying the region where a polynomial system has more than one positive solution is a mathematical problem that belongs to the realm of semi-algebraic geometry and quantifier elimination [15, 16]. Although there are generic methods to address this question, these have high complexity, and fail for realistic networks, even of moderate size. To overcome these difficulties, methods targeting the specificities of reaction networks systems have been developed, and some partial results, for example describing the projection of the multistationarity region onto a subset of the parameters, have been developed [9–11].

Here we present a new algorithm to assert that the multistationarity region is connected without explicitly finding it, which bypasses the use of computationally expensive algorithms from semi-algebraic geometry and quantifier elimination. In fact, we reduce the problem to computing the determinant of a symbolic matrix, and finding a point in the feasible region of a system of linear inequalities. This makes the algorithm successful for networks of moderate size.

We apply our algorithm to numerous motifs in cell signaling, known to exhibit multistationarity, and conclude that these have connected multistationarity regions. These systems are shown in Fig 2, where for all subfigures but (c) and (h), we confirm that the region is connected. For the system in Fig 2(h), previously suggested to have a connected multistationarity region [4], our method is inconclusive. For system (c), modeling the enzymatic phosphorylation of a substrate S with allosteric regulation of the enzymes [17], the algorithm is also inconclusive. In this case, a detailed inspection of the system employing ideas similar to those behind our algorithm, allows us to assert that the region has exactly two path-connected components. These components are contained in the subset of parameters where $\kappa_3 > \kappa_6$ or $\kappa_6 > \kappa_3$ respectively. These two parameters are the catalytic constants of the phosphorylation and dephosphorylation processes, respectively. Therefore, if for example $\kappa_6$ increases from a small value to a value larger than $\kappa_3$, then the multistationarity region will be crossed twice, and a non-simple hysteresis switch arises.

The fact that the remaining networks in Fig 2 have a connected multistationarity region, does not forbid complex switches, as a parameter path could still enter and exit the multistationarity region several times. However, in this scenario we can assert that for any pair of parameter values of multistationarity, there is a path connecting them and completely included in the multistationarity region. This implies that the conditions yielding to multistationarity vary continuously, and we cannot separate the multistationarity region into two disjoint sets, each corresponding to a distinct biological mechanism.

Our algorithm builds on two previous results. First, in [9], the authors associated with each reaction network a function, whose signs are closely related to the multistationarity region. Second, in [18], we developed a new criterion to determine the connectivity of a set described as the preimage of the negative real half-line by a polynomial map. We connect these two key ingredients in Theorem 2 to give a criterion for connectivity of the multistationarity region. Our results establish a stronger property, namely path-connectivity of the region, which in turn imply connectivity.

The criterion relies on computing the determinant of a matrix with symbolic entries, and this task might become unfeasible for large matrices with many variables. To bypass this, we show that the network can be reduced by removing some reactions and connectivity of the multistationarity region for the reduced network can be translated into the original network, see Theorem 4.

This paper is organized as follows. We first establish the framework and background material, and present our algorithm for connectivity of the multistationarity region. We proceed to apply our algorithm to the networks in Fig 2, and while doing so, we illustrate the strengths and limitations of the algorithm. To keep the exposition concise, we compile the proofs of the main theorems in a Proofs section at the end.

## Results

### Theory

The results of this work are framed in the context of chemical reaction network theory, a formalism to study reaction networks that goes back to the 70s with the works of Horn, Jackson and Feinberg [5, 6]. To help the unfamiliar reader, and to fix the notation, we start with a brief introduction. This part ends with the main theoretical result to decide whether the set of parameters where multistationarity arises is path connected (and hence connected). To keep the exposition simple for non-experts, the proofs of the statements are given in the section "Proof of Results" at the end.

**Reaction networks.** A **reaction network** $(\mathcal{S}, \mathcal{R})$ is a collection of *reactions* $\mathcal{R} = \{R_1, \ldots, R_r\}$ between *species* in a set $\mathcal{S} = \{X_1, \ldots, X_n\}$. In our applications, species will be proteins, such as kinases and substrates. The reactions will encode events such as complex formation or posttranslational modifications.

Formally, each reaction connects to linear combinations of species, that is, it has the form:

$$R_j : \ a_{1j}X_1 + \cdots + a_{nj}X_n \longrightarrow b_{1j}X_1 + \cdots + b_{nj}X_n, \ j = 1, \ldots, r,$$

where the coefficients $a_{ij}, b_{ij}$ are non-negative integer numbers. The net production of each species when the reactions takes place is encoded in the **stoichiometric matrix**

$$N = [b_{ij} - a_{ij}]_{\substack{i=1,\ldots,n \\ j=1,\ldots,r}} \in \mathbb{R}^{n \times r}.$$

We do not consider reactions where the reactant and product are equal, hence $N$ has no zero columns.

For illustration purposes, we consider a reaction network that is small enough to get a good feeling about the formal concepts but large enough to display the relevant features. To construct such a reaction network, the article [19] was particularly helpful. Realistic networks will be considered in the **Applications** section of this work. We refer to the following reaction network as the *running example*:

$$X_1 \rightarrow X_2, \ \ X_2 \rightarrow X_1, \ \ 2X_1 + X_2 \rightarrow 3X_1. \tag{1}$$

Here, two species are related by three reactions, so the corresponding stoichiometric matrix has two rows and three columns, see Fig 3.

Mathematical modeling offers us tools to get insights into the dynamics of the network and understand the temporal changes in the concentrations of the species of the network. By encoding the concentrations of $X_1, \ldots, X_n$ into the vector $x = (x_1, \ldots, x_n) \in \mathbb{R}^n_{\geq 0}$, and under the assumption of *mass-action kinetics*, the evolution of the concentrations of the species over

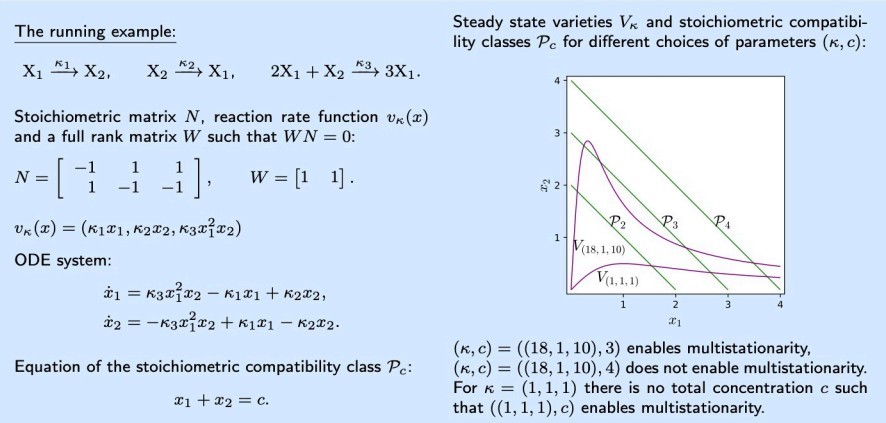

The running example:

$$X_1 \xrightarrow{\kappa_1} X_2, \qquad X_2 \xrightarrow{\kappa_2} X_1, \qquad 2X_1 + X_2 \xrightarrow{\kappa_3} 3X_1.$$

Stoichiometric matrix $N$, reaction rate function $v_\kappa(x)$ and a full rank matrix $W$ such that $WN = 0$:

$$N = \begin{bmatrix} -1 & 1 & 1 \\ 1 & -1 & -1 \end{bmatrix}, \qquad W = \begin{bmatrix} 1 & 1 \end{bmatrix}.$$

$$v_\kappa(x) = (\kappa_1 x_1, \kappa_2 x_2, \kappa_3 x_1^2 x_2)$$

ODE system:

$$\dot{x}_1 = \kappa_3 x_1^2 x_2 - \kappa_1 x_1 + \kappa_2 x_2,$$
$$\dot{x}_2 = -\kappa_3 x_1^2 x_2 + \kappa_1 x_1 - \kappa_2 x_2.$$

Equation of the stoichiometric compatibility class $\mathcal{P}_c$:

$$x_1 + x_2 = c.$$

Steady state varieties $V_\kappa$ and stoichiometric compatibility classes $\mathcal{P}_c$ for different choices of parameters $(\kappa, c)$:

$(\kappa, c) = ((18, 1, 10), 3)$ enables multistationarity, $(\kappa, c) = ((18, 1, 10), 4)$ does not enable multistationarity. For $\kappa = (1, 1, 1)$ there is no total concentration $c$ such that $((1, 1, 1), c)$ enables multistationarity.

**Fig 3. Illustration of the relevant objects for the running example given in (1).**

time is modeled by the ODE system:

$$\dot{x} = f_\kappa(x), \quad x \in \mathbb{R}^n_{\geq 0}, \tag{2}$$

where $f_\kappa(x) := N v_\kappa(x)$ with the *rate function* $v_\kappa(x)$ given for $x \in \mathbb{R}^n_{\geq 0}$ by

$$v_\kappa(x) = \left( \kappa_1 x_1^{a_{11}} \cdots x_n^{a_{n1}}, \ldots, \kappa_n x_1^{a_{1n}} \cdots x_n^{a_{nn}} \right)^\top \in \mathbb{R}^r_{\geq 0}. \tag{3}$$

Here $\kappa = (\kappa_1, \ldots, \kappa_r) \in \mathbb{R}^r_{>0}$ is the vector of *reaction rate constants*, which are parameters of the system. Observe that each component of $v_\kappa(x)$ corresponds to one reaction, and it is obtained by considering the coefficients of the reactant of the reaction as exponents. The function $v_\kappa(x)$ and the associated ODE system of the running example are shown in Fig 3.

The ODE system (2) is forward invariant on **stoichiometric compatibility classes** [20], that is, for any initial condition $x_0$ the dynamics takes place in the stoichiometric compatibility class of $x_0$, which is the set $(x_0 + S) \cap \mathbb{R}^n_{\geq 0}$, where $S$ denotes the vector space spanned by the columns of $N$. To work with stoichiometric compatibility classes it is more convenient to have equations for them. These are obtained by considering a full rank matrix $W \in \mathbb{R}^{(n-s) \times n}$ such that $WN = 0$, where $s$ is the rank of the stoichiometric matrix $N$. Then for each parameter vector $c \in \mathbb{R}^{n-s}$, the associated stoichiometric compatibility class is the set

$$\mathcal{P}_c := \{ x \in \mathbb{R}^n_{\geq 0} \mid Wx = c \}. \tag{4}$$

This class is the set $(x_0 + S) \cap \mathbb{R}^n_{\geq 0}$ for any initial condition $x_0$ satisfying $Wx_0 = c$. Such a matrix $W$ is called a **matrix of conservation relations**, any equation defining a class is a conservation relation, and the parameter vector $c$ is called a *vector of total concentrations*.

For the running example, we have $n = 2$, $s = 1$, and hence $W$ has one row, as given in Fig 3. The figure depicts also the stoichiometric compatibility classes for $c = 2, 3, 4$, which are compact. In general, a reaction network is called **conservative** if each stoichiometric compatibility class is a compact subset of $\mathbb{R}^n_{\geq 0}$, and this is the same as asking that there is a vector with all entries positive in the left-kernel of $N$ [21]. Under this assumption, all trajectories of the ODE system (2) are completely contained in a compact subset, and hence no component can go to infinity. For the purposes of this work, it will be enough to require a milder condition, namely that there is a compact set that all trajectories enter in finite time and do not leave again. In

this case, the reaction network is said to be **dissipative**. See [9] for ways to verify that a network is dissipative. For simplicity, our applications are conservative networks.

We denote the set of non-negative steady states of the ODE system (2) by

$$V_\kappa := \{x \in \mathbb{R}^n_{\geq 0} \mid N v_\kappa(x) = 0\},\tag{5}$$

and call it the **steady state variety**. The **positive steady state variety** consists then of the points in $V_\kappa$ with all entries positive and is denoted by $V_{\kappa,>0}$. A steady state $x \in V_\kappa$ is a **boundary steady state**, if one of its coordinates equals zero. For our results, we will need that stoichiometric compatibility classes intersecting $\mathbb{R}^n_{>0}$ do not contain boundary steady states. We call therefore a boundary steady state **relevant** if it belongs to a class $\mathcal{P}_c$ with $\mathcal{P}_c \cap \mathbb{R}^n_{>0} \neq \emptyset$.

We say that a pair of parameters $(\kappa, c)$ **enables multistationarity** if the intersection of the positive steady state variety $V_{\kappa,>0}$ with the stoichiometric compatibility class $\mathcal{P}_c$ contains more than one point. The set of parameters that enable multistationarity form the **multistationarity region**. We show in the right panel of Fig 3 some stoichiometric compatibility classes, steady state varieties and their intersection, showing that multistationarity arises.

We now recall the main theorem from [9] that is key to describe the multistationarity region. The theorem requires choosing a matrix of conservation relations $W$ that is row reduced. Then, if $i_1 < \cdots < i_{n-s}$ are the indices of the first non-zero coordinates of each row of $W$, we construct the matrix $M_\kappa(x)$ from the Jacobian of $f_\kappa(x)$ by replacing the $i_j$th row by the $j$th row of $W$ (for $j = 1, \ldots, n - s$).

**Theorem 1** (Multistationarity). [9, *Theorem 1*] *Consider a reaction network that is dissipative and does not have relevant boundary steady states. For each vector of reaction rate constants* $\kappa \in \mathbb{R}^r_{>0}$ *and vector of total concentrations* $c \in \mathbb{R}^{n-s}$, *it holds*:

(A) *If* $(-1)^s \det(M_\kappa(x)) > 0$ *for all* $x \in V_{\kappa,>0} \cap \mathcal{P}_c$, *then the parameter pair* $(\kappa, c)$ *does not enable multistationarity.*

(B) *If* $(-1)^s \det(M_\kappa(x)) < 0$ *for some* $x \in V_{\kappa,>0} \cap \mathcal{P}_c$, *then the parameter pair* $(\kappa, c)$ *enables multistationarity.*

Our running example is dissipative, as it is conservative, and it has no relevant boundary steady states. We also find that

$$(-1)^s \det(M_\kappa(x)) = \kappa_3 x_1^2 - 2\kappa_3 x_1 x_2 + \kappa_1 + \kappa_2.$$

This expression is negative for $\kappa = (18, 1, 10)$ and $x^* \approx (0.2448, 2.7552) \in V_\kappa$. Since $Wx^* = 3$, we can conclude using Theorem 1, that the intersection of $V_{(18,1,10)}$ and $\mathcal{P}_3$ contains more than one point. This is exactly what Fig 3 indicates.

**Parametrizations.** In Theorem 1, it is crucial to evaluate the determinant of $M_\kappa(x)$ at points in the incidence set:

$$\mathcal{V} := \{(x, \kappa) \in \mathbb{R}^n_{>0} \times \mathbb{R}^r_{>0} \mid x \in V_\kappa\}.\tag{6}$$

Therefore, we need to be able to describe the points in $\mathcal{V}$ in a useful way. This is done by considering *parametrizations* of $\mathcal{V}$. Loosely speaking, a parametrization is a function whose image set consists precisely of the points in $\mathbb{R}^n_{>0} \times \mathbb{R}^r_{>0}$ that belong to $\mathcal{V}$. Formally, we define a **parametrization of** $\mathcal{V}$ as a surjective analytic map

$$\Phi : \mathcal{D} \to \mathcal{V}.$$

In practice, $\mathcal{D}$ is the positive orthant of some $\mathbb{R}^k$ and $\Phi$ is described by polynomials or

quotients of polynomials such that their denominators do not vanish on $\mathcal{D}$. Below, we discuss how to choose a parametrization and show that there is always at least one.

Using Theorem 1, one can show (see Lemma 9) that the multistationarity region is closely related to the preimage of the negative real half-line under the polynomial map

$$g : \mathcal{V} \to \mathbb{R}, \ (x, \kappa) \mapsto (-1)^s \det(M_\kappa(x)), \tag{7}$$

which can be described using a parametrization

$$g \circ \Phi : \ \mathcal{D} \to \mathcal{V} \to \mathbb{R}, \ \xi \mapsto g(\Phi(\xi)). \tag{8}$$

Following [22], we call the function $g \circ \Phi$ a **critical function**, and observe that it depends on the choice of the parametrization. We are ready to present the main theoretical result of this work, namely a criterion for connectivity of the multistationarity region. The statement tells us that we can look at the number of path-connected components of $(g \circ \Phi)^{-1}(\mathbb{R}_{<0})$, that is, of the set of values $\xi$ where $g \circ \Phi$ is negative. The proof of the following theorem can be found in the Proofs section.

**Theorem 2** (Deciding connectivity). *Consider a reaction network that is dissipative and does not have relevant boundary steady states. Let $g \circ \Phi$ be a critical function as in* (8) *such that the closure of $(g \circ \Phi)^{-1}(\mathbb{R}_{<0})$ equals $(g \circ \Phi)^{-1}(\mathbb{R}_{\leq 0})$.*

*Then the number of path-connected components of the multistationarity region is at most the number of path-connected components of $(g \circ \Phi)^{-1}(\mathbb{R}_{<0})$.*

*In particular, if $(g \circ \Phi)^{-1}(\mathbb{R}_{<0})$ is path connected, then the multistationarity region is path connected.*

## Algorithm for checking path connectivity

Theorem 2 gives a theoretical criterion to decide upon connectivity, from which one can establish an algorithm for connectivity with the following steps:

(Step 1). Check that the reaction network is dissipative and does not have relevant boundary steady states.

(Step 2). Find a parametrization $\Phi$ of $\mathcal{V}$ and compute the critical function $g \circ \Phi$.

(Step 3). Check that $(g \circ \Phi)^{-1}(\mathbb{R}_{<0})$ is path connected and its closure equals $(g \circ \Phi)^{-1}(\mathbb{R}_{\leq 0})$.

The important point is that each of these steps can be addressed computationally, and hence the algorithm can be carried through without manual intervention, at least for networks of moderate size. We proceed to describe each of these steps in detail.

**(Step 1)** has already been described in detail in [9], as it consists of verifying that the conditions to apply Theorem 1 hold. Computable criteria that are sufficient to ensure that the properties hold are presented in [9]. These are, however, not necessary and hence it might not always be possible to decide upon this step.

To verify dissipativity, the first attempt is to show that the reaction network is conservative by finding a row vector $w \in \mathbb{R}^n_{>0}$ such that $wN = 0$. This can be checked by solving the system of linear equalities:

$$w N_i = 0, \ \text{for all} \ i = 1, \ldots, r \ \text{and} \ w_j > 0 \ \text{for all} \ j = 1, \ldots, n, \tag{9}$$

where $N_i$ denotes the $i$th column of $N$. We already noticed that the running example is

conservative, by choosing

$$w = (1, 1) \ \in \mathbb{R}^2_{>0}. \tag{10}$$

A sufficient criterion to preclude the existence of relevant boundary steady states arises by using siphons, that is, subsets of species such that for all species in the set and all reactions producing them, there is a species in the reactant also in the set, see [23, Theorem 2], [24, Proposition 2], [25]. In a nutshell, the criterion requires that for each minimal *siphon* it is possible to choose $w \in \mathbb{R}^n_{\geq 0}$ with $wN = 0$ and such that the positive entries of $w$ correspond exactly to the species in the siphon. For more details, we refer to [9, 23, 24]. Note that this criterion also relies on solving linear inequalities. Our running example has only one siphon, namely $\{X_1, X_2\}$. As the two entries of $w$ in (10) are positive, the criterion holds and the network does not have relevant boundary steady states.

**(Step 2)** asks for the choice of a parametrization of $\mathcal{V}$ and the computation of the critical function. To find a parametrization systematically, we consider so-called *convex parameters* introduced by Clarke in [26]. Since then, they have been applied to study reaction networks, for example to detect Hopf bifurcations and study bistability [27–29].

The idea behind convex parameters is the simple observation that the rate function $v_\kappa(x)$ has to be in the *flux cone*:

$$\mathcal{F} := \left\{ v \in \mathbb{R}^r_{\geq 0} \mid Nv = 0 \right\} = \ker(N) \cap \mathbb{R}^r_{\geq 0}$$

for each $(x, \kappa) \in \mathcal{V}$. It is easy to see that $\mathcal{F}$ is a convex polyhedral cone containing no lines. Using software with packages for polyhedral sets (see Methods), one can compute a minimal collection of generators $E_1, \ldots, E_\ell \in \mathbb{R}^r$ of $\mathcal{F}$.

These generators are often called *extreme vectors* of the cone. Their choice is unique up to multiplication by a positive number. Since the flux cone $\mathcal{F}$ does not contain lines, each of its elements can be written as a non-negative linear combination of *extreme vectors* [30, Corollary 18.5.2], that is for each $v \in \mathcal{F}$ there exists some $\lambda = (\lambda_1, \ldots, \lambda_\ell) \in \mathbb{R}^\ell_{\geq 0}$ such that

$$v = \sum_{i=1}^{\ell} \lambda_i E_i = E\lambda$$

where $E \in \mathbb{R}^{r \times \ell}$ denotes the matrix with columns $E_1, \ldots, E_\ell$. We call $E$ a **matrix of extreme vectors**. This gives rise to the following *convex parametrization*:

$$\Psi : \mathbb{R}^n_{>0} \times \mathbb{R}^\ell_{>0} \rightarrow \mathcal{V}, \ (h, \lambda) \mapsto \ \left( \tfrac{1}{h}, \mathrm{diag}((h^{A_1}, \ldots, h^{A_r}))E\lambda \right), \tag{11}$$

where $A_1, \ldots, A_r$ denote the columns of the matrix $A := [a_{ij}] \in \mathbb{R}^{n \times r}$ of the coefficients of the reactants of the reactions, $h^{A_j}$ is short notation for $h_1^{a_{1j}} \cdots h_n^{a_{nj}}$, $\mathrm{diag}(v)$ is the diagonal matrix with diagonal entries given by $v$, and $1/h$ is taken component-wise.

In Corollary 7(a) in the Proofs section, we show that $\Psi$ is surjective if $E$ does not have a row where all the entries are equal to zero, and hence $\Psi$ is a parametrization of $\mathcal{V}$. This restriction is not relevant for our purposes: a zero row of $E$ is equivalent to $\ker(N)$ not having any positive vector, and hence there is no positive steady state of the ODE system (2), see Corollary 7(b). In particular, the reaction network cannot be multistationary. In the rest of the work, we assume that $E$ does not have a zero row and when this holds, we say that the network is **consistent** [23] (consistent networks are called *dynamically nontrivial* in other works, e.g. [31]).

For the convex parametrization, the critical function $g \circ \Psi$ can be represented in a direct way using the following observation. The Jacobian of $f_\kappa(x)$ evaluated at $\Psi(h, \lambda)$ equals

$$\tilde{J}(h, \lambda) := N\mathrm{diag}(E\lambda)A^\top \mathrm{diag}(h), \tag{12}$$

for each $(h, \lambda) \in \mathbb{R}^n_{>0} \times \mathbb{R}^\ell_{>0}$, see [27]. We construct the matrix $\tilde{M}(h, \lambda)$ from $\tilde{J}(h, \lambda)$ as above: if $W$ is row reduced and $i_1 < \cdots < i_{n-s}$ are the indices of the first non-zero coordinates of each row, replace the $i_j$th row of $\tilde{J}(h, \lambda)$ by the $j$th row of $W$. Then, it holds

$$(g \circ \Psi)(h, \lambda) = (-1)^s \det \tilde{M}(h, \lambda). \tag{13}$$

From this equality, one computes $g \circ \Psi$ directly using symbolic software. Since the entries of $\tilde{M}(h, \lambda)$ are polynomials in $(h, \lambda)$, so is $g \circ \Psi$. In the following, we call this polynomial the **critical polynomial**.

Let us find the critical polynomial $g \circ \Psi$ for the running example. The flux cone $\mathcal{F}$ and its extreme vectors are displayed in Fig 4. Now, all we have to do is to compute the matrix product in (12)

$$\begin{bmatrix} -1 & 1 & 1 \\ 1 & -1 & -1 \end{bmatrix} \begin{bmatrix} \lambda_1 + \lambda_2 & 0 & 0 \\ 0 & \lambda_2 & 0 \\ 0 & 0 & \lambda_1 \end{bmatrix} \begin{bmatrix} 1 & 0 \\ 0 & 1 \\ 2 & 1 \end{bmatrix} \begin{bmatrix} h_1 & 0 \\ 0 & h_2 \end{bmatrix} = \begin{bmatrix} (\lambda_1 - \lambda_2)h_1 & (\lambda_1 + \lambda_2)h_2 \\ -(\lambda_1 - \lambda_2)h_1 & -(\lambda_1 + \lambda_2)h_2 \end{bmatrix},$$

and replace the first row by $(1, 1)$. After taking the determinant and multiplying by $(-1)^s = -1$, we obtain the critical polynomial:

$$(g \circ \Psi)(h_1, h_2, \lambda_1, \lambda_2) = h_1\lambda_2 - h_1\lambda_1 + h_2\lambda_1 + h_2\lambda_2. \tag{14}$$

The above discussion shows that we can always find a suitable parametrization and compute the critical polynomial. In some cases, other types of parametrizations arise by

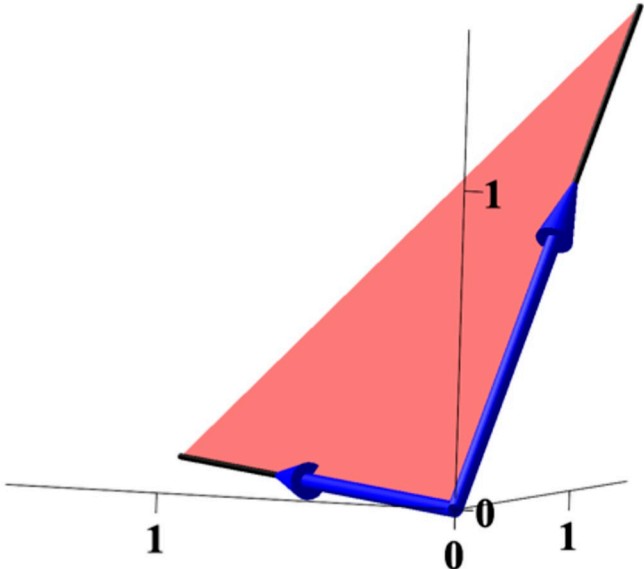

**Fig 4. The flux cone of the running example in $\mathbb{R}^3$.** The extreme vectors $E_1 = (1, 0, 1)$, $E_2 = (1, 1, 0)$ are shown in blue.

parametrizing each $V_{\kappa,>0}$ separately. This is done by first trying to express some variables among $x_1, \ldots, x_n, \kappa_1, \ldots, \kappa_r$ in terms of the others using the equations in (5). For finding these expressions, a computer algebra system such as `SageMath` [32] or `Maple` [33] can be useful. Once such an expression is found, one should check whether it gives a well-defined surjective analytic map, that is a parametrization. If all the components of the parametrization $\Phi$ are quotients of polynomials with positive denominators, then $g \circ \Phi$ is a quotient of polynomials too, and its denominator is positive.

Let us see how this works in practice for the running example. We see from the ODE system in Fig 3 that positive steady states are characterized by

$$x_2 = \frac{\kappa_1 x_1}{\kappa_3 x_1^2 + \kappa_2}.$$

This expression gives the parametrization

$$\Phi : \ \mathbb{R}_{>0}^4 \to \mathcal{V} \subseteq \mathbb{R}_{>0}^2 \times \mathbb{R}_{>0}^3, \ \ (x_1, \kappa_1, \kappa_2, \kappa_3) \mapsto (x_1, \tfrac{\kappa_1 x_1}{\kappa_3 x_1^2 + \kappa_2}, \kappa_1, \kappa_2, \kappa_3).$$

Combining $\Phi$ with $g$ from (7), we get the critical function:

$$(g \circ \Phi)(x_1, \kappa_1, \kappa_2, \kappa_3) = \frac{\kappa_3^2 x_1^4 - \kappa_1 \kappa_3 x_1^2 + 2\kappa_2 \kappa_3 x_1^2 + \kappa_1 \kappa_2 + \kappa_2^2}{\kappa_3 x_1^2 + \kappa_2}. \tag{15}$$

In general, there is no guarantee that such a parametrizations for each $\kappa$ can be found. However, there are broad classes of reaction networks allowing such a parametrization, for example networks with toric steady states [34] and post-translational modification systems [35] to name a few. As we always can find a critical function using the convex parametrization, one might wonder what the value of these other type of parametrizations is. The point is that with this type, the reaction rate constants are still present in the parametrization, and this is useful to get information about what parameter values yield to multistationarity. This was the theme of [9], and we will explore this advantage later in the application of our algorithm to the network with allosteric reciprocal enzyme regulation in Fig 2(c).

Finally, we discuss how to address **(Step 3)**, now that we know how to compute the critical polynomial/function. To check whether the preimage of the negative real half-line under a critical function is path connected is in general hard and depends strongly on the parametrization. As we discussed in (Step 2), critical functions are in practice polynomials or rational functions with positive denominator. In the latter case, we can restrict to the numerator of the rational function. To verify the conditions in Theorem 2, it is then enough to study the preimage of the negative real half-line under a polynomial function restricted to the positive orthant.

Recall that a polynomial function can be written as

$$f : \ \mathbb{R}_{>0}^k \to \mathbb{R}, \ \ f(x) = \sum_{\mu \in \sigma(f)} c_\mu x_1^{\mu_1} \cdots x_k^{\mu_k}, \ \ \text{with } c_\mu \neq 0,$$

and $\sigma(f) \subseteq \mathbb{N}^k$ is a finite set, called the *support* of $f$. To determine whether the *preimage of the negative real half-line*

$$f^{-1}(\mathbb{R}_{<0}) = \{x \in \mathbb{R}_{>0}^k \mid f(x) < 0\}$$

is path connected, one can use methods from real algebraic geometry [15, Remark 11.19], [36, Section 3]. These methods work well for polynomials in few variables, but they scale poorly. If the polynomial has many variables, the computation is unfeasible.

In [18], the authors of the present work gave a sufficient criterion for deciding that $f^{-1}(\mathbb{R}_{<0})$ is path connected, based on the geometry of the support and the sign of the

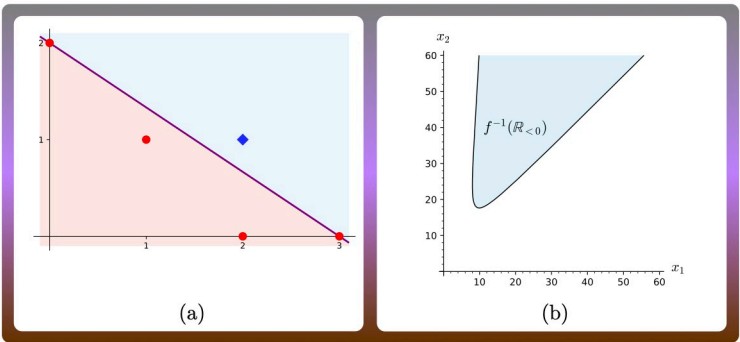

**Fig 5.** (a) A strict separating hyperplane (in purple) of the support of $f(x_1, x_2) = x_1^2 - x_1^2 x_2 + 2x_1 x_2 + x_1^3 + x_2^2$. Red dots correspond to positive exponents, the blue square corresponds to the only negative exponent. (b) The preimage of the negative real half-line under $f$.

coefficients. We call an exponent $\mu \in \sigma(f)$ positive (resp. negative) if the corresponding coefficient $c_\mu$ is positive (resp. negative). We write $\sigma_+(f)$ (resp. $\sigma_-(f)$) for the set of positive (resp. negative) exponents of a polynomial $f$. For example, the polynomial $f(x_1, x_2) = x_1^2 - x_1^2 x_2 + 2x_1 x_2 + x_1^3 + x_2^2$ has four positive exponents $(2, 0)$, $(1, 1)$, $(3, 0)$, $(0, 2)$ and one negative exponent $(2, 1)$. These exponents are depicted in Fig 5.

A hyperplane in $\mathbb{R}^k$ is the set of solutions $\mu \in \mathbb{R}^k$ of a linear equation

$$v \cdot \mu = a,$$

where $v \in \mathbb{R}^k \setminus \{0\}$, $a \in \mathbb{R}$ and $v \cdot \mu$ denotes the Euclidean scalar product of two vectors. Each hyperplane has two sides, which are described by the linear inequalities

$$v \cdot \mu \leq a, \quad \text{and} \quad v \cdot \mu \geq a.$$

A hyperplane is called *strictly separating* if the positive and negative exponents of $f$ are on different sides of the hyperplane and not all the negative exponents are on this hyperplane. For a geometric interpretation, we refer to Fig 5.

Strict separating hyperplanes can be used to decide upon path connectivity of the multistationarity region using Theorem 2 via the following theorem.

**Theorem 3** (Preimage of negative real half-line). [18, *Theorem 3.9*] *Let $f : \mathbb{R}_{>0}^k \to \mathbb{R}$ be a polynomial function. If there exists a strict separating hyperplane of the support of $f$, then $f^{-1}(\mathbb{R}_{<0})$ is path connected and its closure equals $f^{-1}(\mathbb{R}_{\leq 0})$.*

For the running example, the supports of the two critical functions (14) and (15) form a quadrilateral. In both cases, there is only one negative exponent, which is at a corner (vertex) of the quadrilateral. Hence, one can easily find strict separating hyperplanes for the supports of each polynomial. To get a geometric intuition, we investigate the numerator of (15). Its support lives in a 2 dimensional subspace of $\mathbb{R}^4$, so we can project it onto the plane $\mathbb{R}^2$ and find a strict separating hyperplane there. The projected support is precisely that depicted in Fig 5(a).

Therefore, Theorem 3 holds for the running example and any of the two critical functions, and then, by Theorem 2 we conclude that the multistationarity region of the running example is path connected.

Note that a strict separating hyperplane of the support of $f$ exists if the following system of linear inequalities has a solution $(v, a) \in \mathbb{R}^{k+1}$:

$$v \cdot \alpha \leq a, \text{ for all } \alpha \in \sigma_+(f) \tag{16}$$

$$v \cdot \beta \geq a, \text{ for all } \beta \in \sigma_-(f) \tag{17}$$

$$\sum_{\beta \in \sigma_-(f)} (v \cdot \beta - a) > 0. \tag{18}$$

For the polynomial in Fig 5, $v = (2, 3)$, and $a = 6$ form a solution to the system.

In practice, we determine whether the system of linear inequalities (16)–(18) has a solution as follows. First, we construct the polyhedral cone $C \subseteq \mathbb{R}^{k+1}$ defined by the inequalities (16) and (17). Second, we pick a point $(v, a)$ in the relative interior of $C$. If there exists $\beta \in \sigma_-(f)$ such that $v \cdot \beta > a$, then $(v, a)$ satisfies also inequality (18) and a strict separating hyperplane exists. If such $\beta$ does not exist, then a simple argument gives that $\sigma_-(f)$ is contained in the hyperplane defined by any $(w, b) \in C$, i.e. $w \cdot \beta = b$ for all $\beta \in \sigma_-(f)$ and all $(w, b) \in C$. Therefore a strictly separating hyperplane does not exist.

Theorem 3 gives a way to assert that the multistationarity region is path connected, but it is not informative if that is not the case. In [18] additional results are given to include two path-connected components. One of these results will be used to show that the multistationarity region of network in Fig 2(c) has two path-connected components.

**Model reduction for the simplification of the computations.** Finding the critical function or the critical polynomial requires the computation of the determinant of a symbolic matrix, which can have a high computational cost if the matrix is large or the entries are long expressions in the symbolic variables. The next theorem shows that it is possible to remove certain reverse reactions from the network, and use a critical function for the reduced network to study the multistationarity region of the original network, thereby reducing (often dramatically) the computational cost.

**Theorem 4** (Reduction and connectivity). *Consider a conservative reaction network $(\mathcal{S}, \mathcal{R})$ without relevant boundary steady states. Assume that there exist species $X_1, \ldots, X_k \in \mathcal{S}$ such that each $X_j$ participates in exactly 3 reactions of the form*

$$a_{k+1,j} X_{k+1} + \cdots + a_{n,j} X_n \underset{\kappa_{3j}}{\overset{\kappa_{3j-1}}{\rightleftharpoons}} X_j \overset{\kappa_{3j-2}}{\longrightarrow} b_{k+1,j} X_{k+1} + \cdots + b_{n,j} X_n, \; j = 1, \ldots, k.$$

*Let $\tilde{g} \circ \tilde{\Phi}$ be a critical function of the reduced network obtained by removing the reactions corresponding to $\kappa_{3j}$ for $j = 1, \ldots, k$. Assume that the closure of $(\tilde{g} \circ \tilde{\Phi})^{-1}(\mathbb{R}_{<0})$ equals $(\tilde{g} \circ \tilde{\Phi})^{-1}(\mathbb{R}_{\leq 0})$.*

*Then the number of path-connected components of the multistationarity region for both the reduced and the original reaction network is at most the number of path-connected components of $(\tilde{g} \circ \tilde{\Phi})^{-1}(\mathbb{R}_{<0})$.*

*In particular, if $(\tilde{g} \circ \tilde{\Phi})^{-1}(\mathbb{R}_{<0})$ is path connected, then the multistationarity region of the original network $(\mathcal{S}, \mathcal{R})$ is path connected.*

The theorem might look a bit technical, but it is simply saying that it is enough to apply the algorithm to a smaller network obtained by removing the reverse reactions $\kappa_{3j}$, and the conclusions can be translated to the original network. Removal of reverse reactions contribute to the reduction of the computational cost as each of them gives an extreme vector to the flux cone (see Lemma 8(a) in the Proofs section). Making reversible reactions irreversible removes this extreme vector and thereby the matrix $\tilde{M}(h, \lambda)$ depends on one less variable.

To illustrate Theorem 4, we consider the reaction network representing a signaling **cascade with shared kinase** in Fig 2(i). This reaction network describes the phosphorylation of two

substrates $S$ and $P$ with one phosphorylation site. The phosphorylation of $S$ is catalyzed by a kinase $E$, while the phosphorylation of $P$ is catalyzed both by $E$ and by the phosphorylated form of $S$. The dephosphorylation processes are governed by two different phosphatases $F_1$ and $F_2$ [37].

One checks using the above criteria that this network is conservative and has no relevant boundary steady states. Then, Theorem 2 for connectivity of the multistationarity region can be applied. A matrix of extreme vectors, formed by a minimal collection of extreme vectors generating the flux cone is

$$
E = \begin{bmatrix}
0 & 0 & 0 & 0 & 0 & 0 & 0 & 1 \\
0 & 0 & 0 & 0 & 0 & 0 & 1 & 1 \\
0 & 0 & 0 & 0 & 0 & 0 & 1 & 0 \\
0 & 0 & 0 & 0 & 0 & 0 & 0 & 1 \\
0 & 0 & 0 & 0 & 0 & 1 & 0 & 1 \\
0 & 0 & 0 & 0 & 0 & 1 & 0 & 0 \\
1 & 0 & 0 & 0 & 0 & 0 & 0 & 0 \\
1 & 0 & 0 & 1 & 0 & 0 & 0 & 0 \\
0 & 0 & 0 & 1 & 0 & 0 & 0 & 0 \\
0 & 0 & 0 & 0 & 1 & 0 & 0 & 0 \\
0 & 0 & 1 & 0 & 1 & 0 & 0 & 0 \\
0 & 0 & 1 & 0 & 0 & 0 & 0 & 0 \\
1 & 0 & 0 & 0 & 1 & 0 & 0 & 0 \\
1 & 1 & 0 & 0 & 1 & 0 & 0 & 0 \\
0 & 1 & 0 & 0 & 0 & 0 & 0 & 0
\end{bmatrix}
$$

The highlighted column extreme vectors correspond to the 5 reversible reactions of the type in Theorem 4 (the $\kappa_{3j}$ in the theorem). Computing the determinant of (12) takes approximately 1.5 minutes. The critical polynomial has 20 variables and 5312 terms.

Following Theorem 4, we remove the reactions corresponding to $\kappa_3$, $\kappa_6$, $\kappa_9$, $\kappa_{12}$, $\kappa_{15}$. The reduced network has the form

$$
S + E \xrightarrow{\kappa_2} SE \xrightarrow{\kappa_1} S_p + E \qquad S_p + F_1 \xrightarrow{\kappa_5} S_pF_1 \xrightarrow{\kappa_4} S + F_1
$$

$$
P + S_p \xrightarrow{\kappa_8} PS_p \xrightarrow{\kappa_7} P_p + S_p \qquad P + E \xrightarrow{\kappa_{11}} PE \xrightarrow{\kappa_{10}} P_p + E
$$

$$
P_p + F_2 \xrightarrow{\kappa_{14}} P_pF_2 \xrightarrow{\kappa_{13}} P + F_2.
$$

A matrix of extreme vectors is now

$$
\begin{bmatrix}
0 & 0 & 1 \\
0 & 0 & 1 \\
0 & 0 & 1 \\
0 & 0 & 1 \\
1 & 0 & 0 \\
1 & 0 & 0 \\
0 & 1 & 0 \\
0 & 1 & 0 \\
1 & 1 & 0 \\
1 & 1 & 0
\end{bmatrix}.
$$

The extreme vectors are the non-highlighted vectors in the matrix $E$ above, with the entries corresponding to the reverse reactions removed (every third row of $E$). Computing the critical polynomial for the reduced network takes only 4 seconds. This critical polynomial is much simpler than the one for the full network. It has 15 variables and 204 terms.

**Table 1. Summary of the algorithm on selected systems.**

| Reaction network | $n$ | $r$ | $\ell$ | $\#\sigma_+(g \circ \Psi)$ | $\#\sigma_-(g \circ \Psi)$ | sep. hyp. | $\tilde{\ell}$ | $\#\sigma_+(\tilde{g} \circ \tilde{\Psi})$ | $\#\sigma_-(\tilde{g} \circ \tilde{\Psi})$ | sep. hyp. |
|---|---|---|---|---|---|---|---|---|---|---|
| (a) Cell cycle | 6 | 6 | 3 | 5 | 1 | YES | 3 | 5 | 1 | YES |
| (b) Hybrid histidine kinase | 6 | 6 | 2 | 17 | 2 | YES | 2 | 17 | 2 | YES |
| (c) Allosteric regulation | 9 | 10 | 5 | 168 | 8 | NO | 3 | 42 | 2 | NO |
| (d) Covalent regulation | 12 | 14 | 7 | 1856 | 32 | YES | 3 | 116 | 2 | YES |
| (e) 2-site phosph. cycle | 9 | 12 | 6 | 288 | 112 | YES | 2 | 18 | 7 | YES |
| 3-site phosph. cycle | 12 | 18 | 9 | 2560 | 1536 | YES | 3 | 40 | 24 | YES |
| 4-site phosph. cycle | 15 | 24 | 12 | ?? | ?? | ?? | 4 | 75 | 54 | NO |
| (f) 2-site: different phosphat. | 10 | 12 | 6 | 304 | 48 | YES | 2 | 19 | 3 | YES |
| 3-site: diff. phosphat. | 14 | 18 | 9 | 3264 | 960 | YES | 3 | 51 | 15 | YES |
| 4-site: diff. phosphat. | 18 | 24 | 12 | ?? | ?? | ?? | 4 | 127 | 51 | YES |
| (g) 2-site: weak. irrev. dephos. | 11 | 16 | 8 | 2176 | 640 | YES | 4 | 136 | 40 | YES |
| (h) 2-site: fully weak. irrev. | 13 | 20 | 10 | 16320 | 3648 | NO | 6 | 1020 | 228 | NO |
| (i) Two-layer, shared kinase | 12 | 15 | 8 | 5088 | 224 | YES | 3 | 195 | 9 | YES |
| (j) Two layer MAPK cascade | 14 | 18 | 9 | 5120 | 1408 | YES | 3 | 80 | 22 | YES |
| (k) ERK network | 12 | 18 | 9 | 15040 | 3432 | YES | 5 | 1374 | 340 | YES |

The columns of the table indicate: network; $n$ = number of species; $r$ = number of reactions; $\ell$ = number of extreme vectors of the flux cone; $\#\sigma_\pm(g \circ \Psi)$ = number of positive/negative exponents of the critical polynomial; existence of a strict separating hyperplane; $\tilde{\ell}$ = number of extreme vectors of the flux cone of the reduced network of Theorem 4; $\#\sigma_\pm(\tilde{g} \circ \tilde{\Psi})$ = number of positive/negative exponents of the critical polynomial of the reduced network; existence of strict separating hyperplane for the reduced network. The number of variables of the critical polynomial is $n + \ell$ for the original network and $n + \tilde{\ell}$ for the reduced network. ?? means that the computation could not be performed due to computer memory loss. The labels (a)-(k) refer to the networks in Fig 2.

This example illustrates that the reduction in Theorem 4 might reduce the computational cost substantially. On one hand, the computation of the critical polynomial is faster and, on the other, the critical polynomial itself has less variables and terms, and therefore checking **(Step 3)** becomes faster as well. In the next section, we investigate networks where the benefit of applying network reduction and Theorem 4 is more dramatical. For example, for two of the networks, computing the critical polynomial for the full network turned out to be infeasible, but the computation became possible for the reduced network. By means of Theorem 4, we could assert connectivity of the multistationarity region for the full network (see Table 1 for more detail). This illustrates that Theorem 4 allows us to apply our approach to networks that were originally too large.

An important observation is that the existence of a strict separating hyperplane for a network or for a reduced version of it like in Theorem 4 are independent. That is, if we cannot find a strict separating hyperplane for the reduced network, it could still be that it exists for the original network. Also, the existence of this hyperplane depends on the choice of critical function, that is, of the parametrization.

**Algorithm for path connectivity.** We conclude this section by giving a procedure that checks a sufficient criterion for connectivity of the multistationarity region with no user intervention. Since most of the steps rely on solving linear inequalities, we implemented the algorithm using the computer algebra system `SageMath` [32]. The code is given in the Supporting Information file S1 Source Code. We would like to emphasize that the multistationarity region could still be path connected, even if our algorithm terminates inconclusively.

**Algorithm 5**. *Input*: *a reaction network*

*(Step 1) Check that the reaction network is conservative and that it does not have relevant boundary steady states using siphons.*

*(Step 2) Compute the convex parametrization map* Ψ *if the network is consistent, and the critical polynomial g ∘ Ψ from* (13).

*(Step 3) Decide whether a strict separating hyperplane of the support of g ∘ Ψ exists.*

*(Step 4) Eventually repeat Steps 1–3 with a reduced network as in Theorem 4.*

  *Output*: '*The parameter region of multistationarity is path connected*' *or* '*The algorithm is inconclusive*'.

## Investigating connectivity in relevant biochemical networks

We now demonstrate that Algorithm 5 is useful for realistic networks and that the number of connected components of the multistationarity region can be understood for several relevant networks in cell signaling of moderate size.

  We start by going through the algorithm with two small networks: first, with the module regulating the cell cycle shown in Fig 2(a), and then, with the simplified hybrid histidine kinase network in Fig 2(b). The corresponding matrices and the critical polynomials become more complicated than for the running example, but still, are small enough to be displayed here.

  Afterwards we analyze the rest of the networks in Fig 2. Additionally, we consider the extensions of Fig 2(e) and 2(f), with two phosphorylation sites, to several phosphorylation sites, and explore the strengths and weaknesses of the algorithm. When increasing the network size, the computation of the critical polynomial becomes unfeasible and we apply the reduction from Theorem 4.

  We summarize the main properties of all the applications of the algorithm discussed in this work in Table 1. Table 1 shows the number of species, reactions, and extreme vectors of the reaction network. If the critical polynomial can be computed, it shows the number of positive and negative exponents of the critical polynomial and whether a strict separating hyperplane of the support exists. The same computations are repeated with the reduced network of Theorem 4, and we report the same data except the number of species and reactions.

## Small networks

  **Cell cycle regulating module.**   We consider the model proposed in [38] for the second module that regulates the cell's transition from G2 phase to M-phase [39], which is shown in Fig 2(a). This model has been analyzed for bistability in [29]. With the order of species C, $C^+$, M, $M^+$, W, $W^+$, the stoichiometric matrix, a matrix of conservation relations and a matrix of extreme vectors are:

$$N = \begin{bmatrix} 0 & 1 & 0 & 0 & -1 & 0 \\ 0 & -1 & 0 & 0 & 1 & 0 \\ 1 & 0 & -1 & 0 & 0 & 0 \\ -1 & 0 & 1 & 0 & 0 & 0 \\ 0 & 0 & 0 & -1 & 0 & 1 \\ 0 & 0 & 0 & 1 & 0 & -1 \end{bmatrix}, \; W = \begin{bmatrix} 1 & 1 & 0 & 0 & 0 & 0 \\ 0 & 0 & 1 & 1 & 0 & 0 \\ 0 & 0 & 0 & 0 & 1 & 1 \end{bmatrix}, \; E = \begin{bmatrix} 0 & 0 & 1 \\ 0 & 1 & 0 \\ 0 & 0 & 1 \\ 1 & 0 & 0 \\ 0 & 1 & 0 \\ 1 & 0 & 0 \end{bmatrix}.$$

The sum of the three rows of $W$ gives a positive vector and hence the network is conservative. We further verified that the network has no relevant boundary steady states. Furthermore, $E$ has no zero row, so the network is consistent and the critical polynomial can be found using (13). The matrix $\tilde{M}(h, \lambda)$ is found by replacing the first, third and fifth rows of $N \operatorname{diag}(E\lambda)$ $A^\top \operatorname{diag}(h)$ by the rows of $W$:

$$\begin{bmatrix} 1 & 1 & 0 & 0 & 0 & 0 \\ \lambda_2 h_1 & -\lambda_2 h_2 & -\lambda_2 h_3 & 0 & 0 & 0 \\ 0 & 0 & 1 & 1 & 0 & 0 \\ -\lambda_3 h_1 & 0 & \lambda_3 h_3 & -\lambda_3 h_4 & \lambda_3 h_5 & 0 \\ 0 & 0 & 0 & 0 & 1 & 1 \\ 0 & 0 & \lambda_1 h_3 & 0 & \lambda_1 h_5 & -\lambda_1 h_6 \end{bmatrix}.$$

Since $s = 3$, the negative of the determinant of $\tilde{M}(h, \lambda)$ gives the critical polynomial:

$$(g \circ \Psi)(h, \lambda) = (-h_1 h_3 h_5 + h_1 h_4 h_5 + h_2 h_4 h_5 + h_2 h_3 h_6 + h_1 h_4 h_6 + h_2 h_4 h_6)\lambda_1 \lambda_2 \lambda_3.$$

A strict separating hyperplane exists, for example $v \cdot (h_1, \ldots, h_6, \lambda_1, \lambda_2, \lambda_3) = 2$ with

$$v = (1, 0, 1, 0, 1, 0, 0, 0, 0).$$

Indeed, $(g \circ \Psi)(h, \lambda)$ has 6 monomials, all with exponent $(1, 1, 1)$ for $\lambda$, and for $h$ they have exponents $(1, 0, 1, 0, 1, 0)$, $(1, 0, 0, 1, 1, 0)$, $(0, 1, 0, 1, 1, 0)$, $(0, 1, 1, 0, 0, 1)$, $(1, 0, 0, 1, 0, 1)$, $(0, 1, 0, 1, 0, 1)$. The first exponent is negative, and the scalar product with $v$ returns the value 3, which is strictly larger than 2. The other exponents correspond to positive coefficients, and their scalar product with $v$ give the values 2, 1, 1, 1, 0 respectively. All of them are smaller or equal to 2. Therefore, the condition for being a strict separating hyperplane holds, and we conclude that the multistationarity region of this network is path connected.

**Hybrid histidine kinase.** The hybrid histidine kinase network in Fig 2(b) comprises a hybrid histidine kinase HK with the domain REC embedded, and separate histidine phosphotransfer domain Hpt. This reaction network has been studied in [40], where it was shown that the network displays multistationarity and a (complicate) description of the set of parameters with 3 steady states is given. It is further known that there is a choice of total concentrations such that the network is multistationary if and only if $\kappa_3 > \kappa_1$, see [9] (this set is the projection of the multistationarity region onto the space of reaction rate constants). It was not known whether the full multistationarity region in $\kappa$ and $c$ is path connected.

With the order of species $HK_{00}$, $HKp_0$, $HK_{0p}$, $HK_{pp}$, $Hpt$, $Hpt_p$, the stoichiometric matrix $N$, a matrix of conservation relations $W$, and a matrix $E$ whose columns are a minimal set of extreme vectors are

$$N = \begin{bmatrix} -1 & 0 & 0 & 1 & 0 & 0 \\ 1 & -1 & 0 & 0 & 1 & 0 \\ 0 & 1 & -1 & -1 & 0 & 0 \\ 0 & 0 & 1 & 0 & -1 & 0 \\ 0 & 0 & 0 & -1 & -1 & 1 \\ 0 & 0 & 0 & 1 & 1 & -1 \end{bmatrix}, \quad W = \begin{bmatrix} 1 & 1 & 1 & 1 & 0 & 0 \\ 0 & 0 & 0 & 0 & 1 & 1 \end{bmatrix}, \quad E = \begin{bmatrix} 0 & 1 \\ 1 & 1 \\ 1 & 0 \\ 0 & 1 \\ 1 & 0 \\ 1 & 1 \end{bmatrix}.$$

The network is conservative, consistent, and has no relevant boundary steady states. By replacing the first and fifth rows of $N \operatorname{diag}(E\lambda)A^\top \operatorname{diag}(h)$ by the rows of $W$ we find the matrix $\tilde{M}(h, \lambda)$ and its determinant gives the critical polynomial:

$$
\begin{aligned}
(g \circ \Psi)(h, \lambda) &= h_1 h_2 h_3 h_5 \lambda_1^3 \lambda_2 + 3\, h_1 h_2 h_3 h_5 \lambda_1^2 \lambda_2^2 + 2\, h_1 h_2 h_3 h_5 \lambda_1 \lambda_2^3 + h_1 h_2 h_4 h_5 \lambda_1^2 \lambda_2^2 \\
&\quad + h_1 h_2 h_4 h_5 \lambda_1 \lambda_2^3 - h_2 h_3 h_4 h_5 \lambda_1^3 \lambda_2 - h_2 h_3 h_4 h_5 \lambda_1^2 \lambda_2^2 + h_1 h_2 h_3 h_6 \lambda_1^3 \lambda_2 \\
&\quad + 2\, h_1 h_2 h_3 h_6 \lambda_1^2 \lambda_2^2 + h_1 h_2 h_3 h_6 \lambda_1 \lambda_2^3 + h_1 h_2 h_4 h_6 \lambda_1^3 \lambda_2 + 2\, h_1 h_2 h_4 h_6 \lambda_1^2 \lambda_2^2 \\
&\quad + h_1 h_2 h_4 h_6 \lambda_1 \lambda_2^3 + h_1 h_3 h_4 h_6 \lambda_1^3 \lambda_2 + 2\, h_1 h_3 h_4 h_6 \lambda_1^2 \lambda_2^2 + h_1 h_3 h_4 h_6 \lambda_1 \lambda_2^3 \\
&\quad + h_2 h_3 h_4 h_6 \lambda_1^3 \lambda_2 + 2\, h_2 h_3 h_4 h_6 \lambda_1^2 \lambda_2^2 + h_2 h_3 h_4 h_6 \lambda_1 \lambda_2^3.
\end{aligned}
$$

The polynomial has 8 variables and 17 positive and 2 negative coefficients. A strict separating hyperplane of its support is given by the equation

$$
(-5, -5, -1, 0, 5, 0, 0, 0) \cdot \mu = -3.
$$

using Theorem 2, it follows that the multistationarity region for the hybrid histidine kinase network is path connected.

## Phosphorylation cycles

We investigate models for phosphorylation and dephosphorylation of a substrate $S$ with $m$ binding sites with processes catalyzed by a kinase $E$ and one or more phosphatases $F$.

**Sequential and distributive phosphorylation cycles.** We first assume that phosphorylation and dephosphorylation occurs sequentially and distributively [41]: the kinase $E$ catalyzes the phosphorylation one site at a time in a given order, while the phosphatase $F$ dephosphorylates in the reverse order, also one site at a time. Under these assumptions, the network for $m = 2$ sites is shown in Fig 2(e).

The dynamics of phosphorylation cycles have been intensively studied, e.g. [10, 14, 27, 41–50]. In particular, it is known that they are multistationary for $m \geq 2$, and there are choices of parameter values where they have $m + 1$ steady states for $m$ even, and $m$ for $m$ odd [43], with half of them plus one being asymptotically stable [46]. It has been conjectured that these networks can have up to $2m - 1$ steady states, but this has only been established for small $m$ [47]. These networks are in the class of post-translational modification networks, which are conservative and consistent, and by the results in [24], since the underlying substrate network is strongly path connected, they do not have relevant boundary steady states.

The phosphorylation cycle with $m = 2$ binding sites has $n = 9$ species and the flux cone has $\ell = 6$ extreme vectors. Then, the corresponding critical polynomial $g \circ \Psi$ has 15 variables. It is a big polynomial with 288 positive and 112 negative exponents. Despite the large number of exponents, a strict separating hyperplane of its support can be found in less than a second. Our algorithm (and Theorem 2) can be applied to conclude that the multistationarity region for the sequential and distributive phosphorylation cycle with two binding sites is path connected.

By increasing the number $m$ of binding sites, the reaction network size increases systematically. For example, for $m = 3$, the reaction network becomes

$$
S + E \underset{\kappa_2}{\overset{\kappa_1}{\rightleftarrows}} ES \xrightarrow{\kappa_3} S_p + E \underset{\kappa_8}{\overset{\kappa_7}{\rightleftarrows}} ES_p \xrightarrow{\kappa_9} S_2 + E \underset{\kappa_{14}}{\overset{\kappa_{13}}{\rightleftarrows}} ES_{pp} \xrightarrow{\kappa_{15}} S_{ppp} + E
$$

$$
S_{ppp} + F \underset{\kappa_{17}}{\overset{\kappa_{16}}{\rightleftarrows}} FS_{ppp} \xrightarrow{\kappa_{18}} S_{pp} + F \underset{\kappa_{11}}{\overset{\kappa_{10}}{\rightleftarrows}} FS_{pp} \xrightarrow{\kappa_{12}} S_p + F \underset{\kappa_5}{\overset{\kappa_4}{\rightleftarrows}} FS_p \xrightarrow{\kappa_6} S + F.
$$

The critical polynomial $g \circ \Psi$ has 2560 positive and 1536 negative exponents. The algorithm confirms that the multistationarity region is path connected in 96 seconds.

For $m = 4$, we could not compute the critical polynomial for the original network due to computer memory constraints. This shows that the computation of the critical polynomial is the bottleneck of the algorithm. After removing all reverse reactions as in Theorem 4, we could compute the critical polynomial, but it does not have a strict separating hyperplane. Therefore, for this family of networks we know that the multistationarity region is path connected for $m = 2, 3$, but it is unknown for $m \geq 4$. Previous work has shown that the projection of the multistationarity region onto the reaction rate constants $\kappa$ is path connected for all $m \geq 2$, see [51], so we conjecture that the full multistationarity region is path connected for all $m \geq 2$.

**Phosphorylation cycles with different phosphatases.** Different mechanisms for multisite phosphorylation have been observed, and in particular, phosphorylation and dephosphorylation of the different sites of a phosphate might not be catalyzed by the same kinase or phosphatase e.g. [52, 53]. If all steps are carried out by different enzymes, then multistationarity does not arise (see [37] for $m = 2$). Therefore, we consider the scenario where the phosphorylation occurs sequentially and the kinase acts in a distributive way, but we assume that each dephosphorylation step is governed by **different phosphatases** $F_1, \ldots, F_m$ (see Fig 2(f) for $m = 2$). These networks are also conservative and do not have relevant boundary steady states for any $m$.

For $m = 2$ and $m = 3$, the algorithm finds a strict separating hyperplane, and hence the multistationarity region is path connected. For $m = 4$, the computation of the critical polynomial via the symbolic determinant in (Step 2) of the algorithm was too demanding, and the computer used for the tests ran out of memory. We employed the reduction approach given in Theorem 4 and removed eight reactions. The critical polynomial of the reduced network is significantly simpler: it has 22 variables and 178 monomials. Its support has a strict separating hyperplane. Thus, the multistationarity region of the original network is path connected by Theorem 4 and Theorem 3.

**Weakly irreversible phosphorylation cycles.** The two-site sequential and distributive phosphorylation network given in Fig 2(e) assumes that each phosphorylation step proceeds via a Michaelis-Menten mechanism. This is referred to as *strong irreversibility* in [42]. More plausible mechanisms have been argued to include the complex formation of the product with the enzyme, as for example a mechanism of this form would allow:

$$S + E \rightleftharpoons SE \rightarrow S_p E \rightleftharpoons S_p + E.$$

A model incorporating this **weak irreversibility** at the dephosphorylation stage was proposed and analyzed for bistability in [54, Scheme 2]. In the model, dephosphorylation of ERK by the phosphatase MKP3 proceeds as shown in Fig 2(g) [55]. For this network, our algorithm concludes that the multistationarity region is path connected.

A model with **full weak irreversibility**, that is, for both the phosphorylation and dephosphorylation processes, is shown in Fig 2(h). The shape of the multistationarity region for this type of models was analyzed in [42], where it was concluded by means of a numerical approach that the multistationarity region in some aggregated steady state parameters is connected. Neither for the original network nor for the reduced network, a strict separating hyperplane exists, and hence our algorithm is inconclusive. It remains thus open to be confirmed whether the multistationarity region is path connected.

**Extracellular signal-regulated kinase (ERK) network.** Dual-site phosphorylation and dephosphorylation of extracellular signal-regulated kinase has an important role in the

regulation of many cellular activities [56], and a better knowledge of the dynamical properties of the ERK network might facilitate the prediction of this network's response to environmental changes or drug treatments [57]. This network, analyzed in [58–60] and shown in Fig 2(k), comprises as well phosphorylation of a substrate in two sites, but not in a distributive and sequential way.

Using Algorithm 5, we conclude that the multistationarity region for the ERK network is path connected.

## Signaling cascades

We next investigate two types of signaling cascades comprising phosphorylation cycles, which are known to be multistationary.

**Shared kinase.** We consider first a two-layer signaling cascade with two single phosphorylation at each stage. The phosphorylated substrate of the first layer acts as the kinase of the second layer. We consider additionally that the kinase of the first layer also can act as kinase for the second layer, so the kinase is *shared* for the two layers, as shown in Fig 2(i). Without this shared kinase, the cascade would not display multistationarity.

Algorithm 5 finds a strict separating hyperplane and we conclude that the multistationarity region for the cascade is path connected.

**Two-layer MAPK-cascade.** Huang and Ferrell proposed a model for the MAPK cascade consisting of three layers, the first one being a single phosphorylation cycle, while the last two are dual phosphorylation cycles with phosphorylation and dephosphorylation proceeding sequentially and distributive [61]. This network has bistability and also oscillations, and in fact for both properties only the first two layers of the cascade are required.

The network with two layers is shown in Fig 2(j), and Algorithm 5 can be employed to conclude that the multistationarity region is path connected.

The full network with the three layers is large, with 22 species, 30 reactions, the rank of the stoichiometric matrix is 15, and the matrix $E$ has 15 extreme vectors. Due to the computational cost, the computation of the critical polynomial was not possible for the original network, but was possible for the reduced network using the `Julia` package `SymbolicCRN.jl` [62]. However, a strict separating hyperplane does not exist, so our algorithm is inconclusive.

## Reciprocal enzyme regulation

Finally, we consider two multistationary networks comprising single phosphorylation cycles, where the kinase and the phosphatase are subject to reciprocal regulation, both proposed and studied in [17].

**Covalent regulation.** We first consider the case where reciprocal regulation is via covalent modification catalyzed by the same enzyme, see Fig 2(d). By means of Algorithm 5 we find that the multistationarity region for the the cascade is path connected.

**Allosteric regulation: An example with two path-connected components.** The other mechanism of reciprocal regulation considered in [17] is via allosteric regulation: it is assumed that there is an allosteric effector L that binds both the phosphatase and the kinase, see Fig 2(c). After performing a quasi-steady-state approximation, the authors of [17] show that a necessary condition for multistationarity is that $\kappa_3 > \kappa_6$. This network is conservative, consistent, and has no relevant boundary steady states.

For all the applications we have seen so far, we could either conclude that the multistationarity region is path connected, or our approach was inconclusive. For this network our approach was inconclusive as well, however, by employing other theoretical results from [18]

in conjunction with Theorem 4 and the approaches in [9], we were able to conclude that the multistationarity region has exactly two path-connected components, revealing two mechanisms underlying multistationarity.

The approach is as follows. On one hand, using the method to find parameter regions in $\kappa$ from [9] relying on Theorem 1, we show in Proofs section that the two sets of parameters $\{(\kappa, c) \in \mathbb{R}^{10}_{>0} \times \mathbb{R}^{5}_{\geq 0} \,|\, \kappa_3 > \kappa_6\}$ and $\{(\kappa, c) \in \mathbb{R}^{10}_{>0} \times \mathbb{R}^{5}_{\geq 0} \,|\, \kappa_3 < \kappa_6\}$ both contain parameters that yield multistationarity, that is, these two regions both intersect the multistationarity region. To be precise, the condition $\kappa_3 > \kappa_6$ yields multistationarity if additionally the Michaelis-Menten constant for phosphorylation, $K_1 = \frac{\kappa_2 + \kappa_3}{\kappa_1}$, is large enough relative to that for the dephosphorylation, $K_2 = \frac{\kappa_5 + \kappa_6}{\kappa_4}$. Symmetrically, the condition $\kappa_6 > \kappa_3$ requires $K_2 \gg K_1$ for multistationarity to arise.

We also show that if $\kappa_3 = \kappa_6$, then the network cannot display multistationarity, no matter what the other parameters are. Therefore, any two points in each of the two sets above cannot be joined by a continuous path inside the multistationarity region, as any such path should cross at least one point where $\kappa_3 = \kappa_6$. Hence, the full multistationarity region cannot be path connected: it has *at least* two path-connected components.

On the other hand, we show also in Proofs section that the multistationarity region of the reduced network has *at most* two path-connected components using [18]. Hence, by Theorem 4, the original network in Fig 2(c) has at most two path-connected components as well.

Putting the two pieces together, we conclude that the multistationarity region has precisely two connected components: one in which the catalytic rate of the phosphorylation step is larger than the catalytic rate of dephosphorylation, that is $\kappa_3 > \kappa_6$, and the other where the inequality is reversed $\kappa_6 > \kappa_3$. The second region was missed in [17] as it is outside the regime where the quasi-steady-state approximation employed there is valid.

To illustrate the type of switches that may arise from this system, we have considered the reduced model (for which there are also two connected components), and selected two parameter values $\alpha_1$, $\alpha_2$ in different path-connected components of the multistationarity region. The two parameter values are identical except for the three parameters governing the dephosphorylation event: $\kappa_4$, $\kappa_6$ and the total amount of P, see Fig 6. We choose the path through the two points, which component-wise is given as $\beta(t)_i = \alpha_{1,i}^t \alpha_{2,i}^{1-t}$. At $t = 0$ the path is $\alpha_1$, while at $t = 1$ it is $\alpha_2$. Fig 6(a) and 6(c) shows bifurcation diagrams, where $t$ is the perturbed parameter, which perturbs simultaneously $K_2$, $\kappa_6$ and the total amount of P, and we display the logarithm of the concentration of the phosphorylated substrate $S_p$, kinase K and ligand L at steady state respectively. Note that by the choice of path we have made, $t$ can take any real value, also negative. For the three concentrations, we obtain saddle-node bifurcations: a usual switch for larger values of $t$, while for smaller values of $t$, no hysteresis effect arises and the response curve has two components. In panel (d) of Fig 6 a path in the three dimensional parameter space joining the two points is given. We see that the path enters and exists the multistationarity region twice, corresponding to the two path-connected components. The shape of these two components is displayed in Fig 6(e) and 6(f) after slicing the three-dimensional space by fixing $\kappa_6$ for illustration purposes.

## Discussion

Determining topological properties of semi-algebraic sets, that is, sets described by polynomial equalities and inequalities, is a highly complex problem that requires, for general sets, computationally expensive algorithms that scale poorly with the number of variables [15]. For

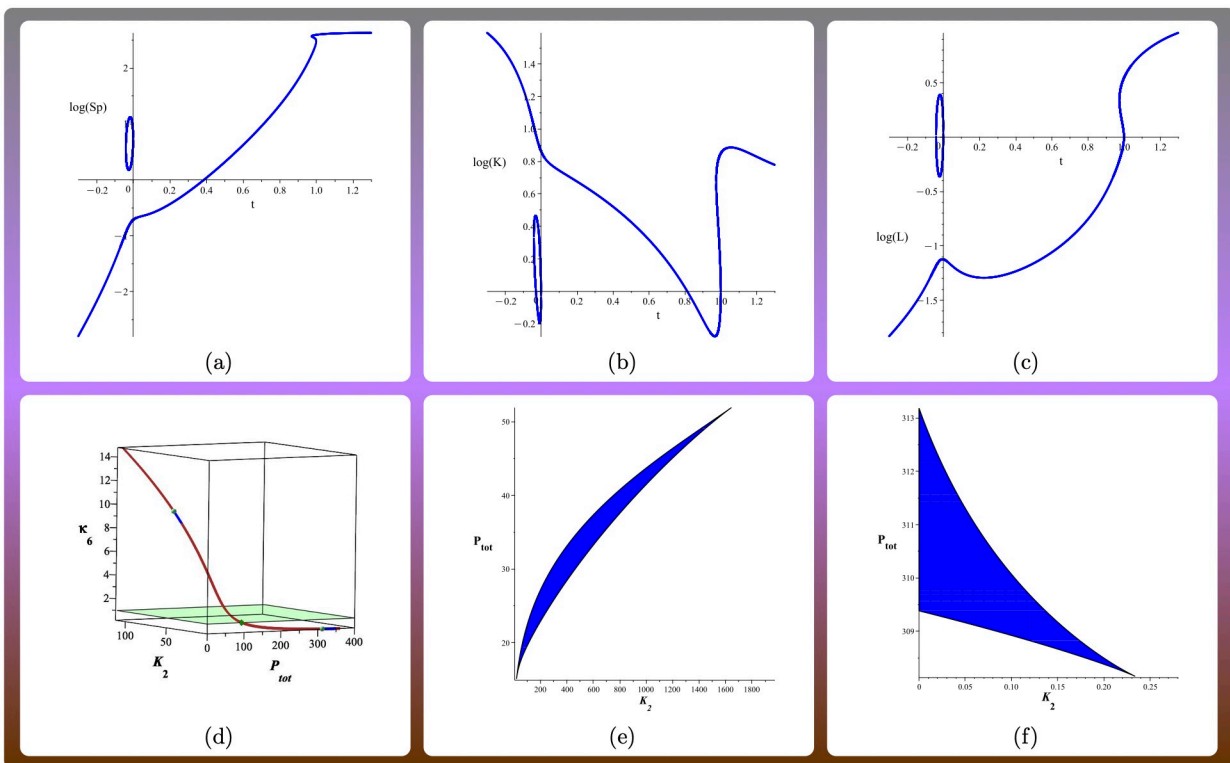

**Fig 6. Input-output curves (bifurcation diagrams) for the reduced reciprocal allosteric regulation system in Fig 2(c) (that is, with $\kappa_2 = \kappa_5 = 0$ for simplicity).** The following parameters are fixed: $\kappa_1 = 1$, $\kappa_3 = 1$, $\kappa_7 = 1$, $\kappa_8 = 1$, $\kappa_9 = 1$, $\kappa_{10} = 0.1$, $L_{tot} = 72$, $K_{tot} = 62$, $S_{tot} = 426$. In (a)-(d), the bifurcation parameter $t$, which can be negative, describes a path for $(\kappa_4, \kappa_6, P_{tot}) : \beta(t) = \left( \left(\frac{1}{5}\right)^t \left(\frac{510}{41}\right)^{1-t}, 10^t \left(\frac{51}{256}\right)^{1-t}, 17^t \left(\frac{5307}{17}\right)^{1-t} \right)$. Subfigures (a)-(c) show the bifurcation diagrams for the concentration of $S_p$, K and L at steady state. In the intervals with three steady states, the one in the middle is unstable. Subfigure (d) shows the path in the three-dimensional space $(K_2, P_{tot}, \kappa_6)$. The blue regions indicate the region of the path that belongs to the multistationarity region, and the displayed plane $\kappa_6 = 1$ separates the two regions. Subfigures (e)-(f) show the multistationarity regions when we fix $\kappa_6 = 9.5$ and $\kappa_6 = 0.195$ respectively. These are two slices of the two path-connected components, obtained by keeping only two free parameters.

reaction networks with mass-action kinetics, the multistationarity region is a semi-algebraic set, and hence its description might not be straightforward.

Here we presented an approach to determine a basic topological property of a set, namely its connectivity. Non-connectivity of the multistationarity region may indicate different biological mechanisms underlying the existence of multiple steady states, and additionally, may give the cell the possibility to operate on complex switches as shown in Fig 1. Our algorithm is to our knowledge the first to address the problem of connectivity in an effective and conclusive way. This is done by relying on linear programming and polyhedral geometry algorithms, rather than on semi-algebraic approaches, which reduces dramatically the computational cost. Additionally, our approach provides a symbolic proof of connectivity, and does not require numerical approaches, which unavoidably cannot explore the whole parameter space.

Although our algorithm might terminate inconclusively, even if the multistationarity region is path connected, we have shown that it is often applicable: for many motifs, the multistationarity region is connected because a strict separating hyperplane of the support of the critical polynomial exists. This came as a surprise to us, and might indicate a hidden feature present in

realistic systems and brings up the question: What are the characteristics of the reaction networks from cell signaling that ensure that the support of the corresponding critical polynomial has a strict separating hyperplane?

It would certainly be relevant to understand the answer to this question, to bypass finding the critical polynomial, a step that is prohibitive for larger networks. This was illustrated with the networks of phosphorylation cycles with several phosphorylation sites, which revealed the computational boundaries of the algorithm. In several cases, the computation of the critical polynomial was not possible on a common computer. However, it was possible to compute the critical polynomial of the reduced network and still we were able to conclude that the multistationarity region is path connected.

Despite covering many networks, the algorithm remains inconclusive for some relevant networks, where we cannot decide whether the multistationarity region is connected, meaning that further investigations are required. For the $m$-site sequential and distributive systems, the projection of the multistationarity region onto the set of reaction rate constants is known to be path connected for all $m$ [51], and this makes us believe that the same holds for the full region. In fact, based on the evidence gathered from the tested networks, we conjecture that if the projection onto the set of reaction rate constants $\kappa$ is path connected, so is the multistationarity region. This would provide an additional strategy to study connectivity, which in particular might give a way to show that the fully weakly irreversible phosphorylation cycle studied in [4], see Fig 2(h), is indeed path connected.

For the network in Fig 2(c), where the algorithm was inconclusive, the network had two path-connected components. The strategy to show this was to combine knowledge about the projection of the multistationarity region onto the set of reaction rate constants, and a bound on the number of connected components of the reduced network found using ideas similar to those in the proof of [18, Theorem 3.9]. This example opens up for new directions for counting path-connected components and understanding underlying features of reaction networks where the multistationarity region is disconnected. On one hand, it would be interesting to devise algorithms that can assert that the multistationarity region is disconnected, and ideally, count or give bounds on the number of path-connected components. On the other hand, one might wonder what network characteristics might give rise to disconnected multistationarity regions. For the reduced network of Fig 2(c), the multistationarity region is no longer disconnected after deleting a reaction or a species, so this network can be viewed as a minimal motif with this property. A proper investigation of minimal networks with disconnected multistationarity region would require a better understanding on how the connectivity of the multistationarity region changes upon modifications on the network, in the spirit of Theorem 4.

We would like to point out that the proof of the key theorem, namely Theorem 2, is based on relating the multistationarity region to the preimage of the negative real half-line by the critical polynomial. When a strict separating hyperplane of the support of the critical polynomial exists, it is not only known that this preimage is path connected, but also that it is contractible [18, Theorem 3.9]. This implies that all Betti numbers of the preimage set are zero. We conjecture that when this is the case, the multistationarity region is contractible, and hence topologically very simple (for example, it has no holes). However, this cannot be directly deduced by our arguments in the proof of Theorem 2.

To conclude, we propose the application of our work in the design of synthetic circuits displaying predefined switches. Indeed, by combining algorithms to determine multistationarity with the study of the connectivity of the multistationarity region, one can systematically study small networks and search for a desired input-output curve shape. This approach would adhere to related work already done in this direction, e.g. [63, 64].

## Methods

We implemented Algorithm 5 in `SageMath` 9.2 [32]. A Jupyter notebook containing the code can be found in the Supporting Information file S1 Source Code. The computations for the networks in Table 1 were run on a Windows 10 computer with Intel Core i5–10310U CPU @ 1.70GHz 2.21 GHz processor and 8GB RAM.

In our implementation, each species is represented by a *symbolic variable*, and each reaction is represented by a list containing two *symbolic expressions* in the variables. For example, to run the code for the running example (1), one has to type:

```
X1,X2 = var('X1,X2')
species = [X1,X2]
reactions = [[X1,X2],[X2,X1],[2*X1+X2,3*X1]]
F = CheckConnectivity(species,reactions)
```

The output is given in the following format:

```
n = 2
r = 3
The reaction network is conservative.
There are no relevant boundary steady states.
l = 2
Number of positive coefficients: 3
Number of negative coefficients: 1
The support set has a strict separating hyperplane.
All the conditions are satisfied.
We conclude that the parameter region of multistationarity
is path connected.
```

Although we used `SageMath` to compute the extreme vectors, the same computation can also be done with `Polymake` [65] (as a standalone [66] or as part of the `Oscar` project in `Julia` [67]). These programs compute extreme vectors of arbitrary pointed cones. For open cones of the form $\ker(N) \cap \mathbb{R}_{>0}^n$, there exist specific algorithms designed in the context of stoichiometric network analysis and metabolic network analysis. See for example [68].

As discussed above, the bottleneck of Algorithm 5 is to compute the determinant of the symbolic matrix $\tilde{M}(h, \lambda)$ in (13). In the tested examples, the `Julia` package `SymbolicCRN.jl` [62] is more efficient in computing the symbolic determinant than our implementation using `SageMath`. Using `SymbolicCRN.jl`, we were able to compute the critical polynomial for the reduced MAPK cascade with three layers, which was not possible with `SageMath`. For the 4-site phosphorylation networks in Table 1, we could not compute the critical polynomial using neither `Julia` nor `SageMath`.

The parameter regions of multistationarity displayed in Fig 6(e) and 6(f) are found using cylindrical algebraic decomposition in `Maple`, using the command `CellDecomposition` from the packages `RootFinding[Parametric]` and `RegularChains`.

## Proof of the results

### Convex parameters

In this subsection we show basic results on the flux cone, and specially that positive combinations of extreme vectors of $\ker(N) \cap \mathbb{R}_{\geq 0}^r$ parametrize the positive part of the cone.

**Proposition 6**. *Let $E \in \mathbb{R}^{r \times \ell}$ be a matrix of extreme vectors for the flux cone* $\ker(N) \cap \mathbb{R}^r_{\geq 0}$.

*(a) The relative interior of* $\ker(N) \cap \mathbb{R}^r_{\geq 0}$ *equals* $\{E\lambda \mid \lambda \in \mathbb{R}^\ell_{>0}\}$ *and contains* $\ker(N) \cap \mathbb{R}^r_{>0}$.

*(b)* $\ker(N) \cap \mathbb{R}^r_{>0} \neq \emptyset$ *if and only if E does not have any zero row.*

*(c) If E does not have any zero row, then* $\{E\lambda \mid \lambda \in \mathbb{R}^\ell_{>0}\} = \ker(N) \cap \mathbb{R}^r_{>0}$.

*Proof.* (a) The first equality is proven in [69, Section XVIII, Theorem 1]. To show that $\ker(N) \cap \mathbb{R}^r_{>0}$ is included in $\{E\lambda \mid \lambda \in \mathbb{R}^\ell_{>0}\}$, let $v \in \ker(N) \cap \mathbb{R}^r_{>0}$ and $\mathbf{1} \in \mathbb{R}^\ell$ the vector with coordinates equal to 1. By [30, Corollary 18.5.2], there exists $\lambda \in \mathbb{R}^\ell_{\geq 0}$ such that $v = E\lambda$. Since all coordinates of $v$ are positive, it is possible to choose $\epsilon > 0$ such that $E\lambda - \epsilon E \mathbf{1}$ has positive coordinates. Thus, $E(\lambda - \epsilon \mathbf{1})$ belongs to the flux cone. Using [30, Corollary 18.5.2] again, there exist $\mu \in \mathbb{R}^\ell_{\geq 0}$ such that $E(\lambda - \epsilon \mathbf{1}) = E\mu$. By reordering, we have

$$v = E\lambda = E(\mu + \epsilon \mathbf{1})$$

where $\mu + \epsilon \mathbf{1} \in \mathbb{R}^\ell_{>0}$. This shows (a).

(b) Follows easily from the equality $\ker(N) \cap \mathbb{R}^r_{\geq 0} = \{E\lambda \mid \lambda \in \mathbb{R}^\ell_{\geq 0}\}$.

(c) If $E$ does not have a zero row, then $\{E\lambda \mid \lambda \in \mathbb{R}^\ell_{>0}\} \subseteq \ker(N) \cap \mathbb{R}^r_{>0}$. Together with (a), we obtain the equality of sets.

**Corollary 7**. *Let $E \in \mathbb{R}^{r \times \ell}$ be a matrix of extreme vectors for the flux cone* $\ker(N) \cap \mathbb{R}^r_{\geq 0}$. *Recall the set $\mathcal{V}$ of steady states given in* (6).

*(a) If E does not have any zero row, then the convex parametrization map* $\Psi : \mathbb{R}^n_{>0} \times \mathbb{R}^\ell_{>0} \to \mathcal{V}$ *from* (11) *is surjective.*

*(b) If E has a zero row, then* $\mathcal{V} = \emptyset$.

*Proof.* First, we observe that for $(x, \kappa) \in \mathcal{V}$ the vector $v_\kappa(x)$ lies in $\ker(N) \cap \mathbb{R}^r_{>0}$ by (3). Now, part (b) follows directly from Proposition 6(b). To show that $\Psi$ is surjective, let $(x, \kappa) \in \mathcal{V}$. By Proposition 6(a), there exist $\lambda \in \mathbb{R}^\ell_{>0}$ such that $v_\kappa(x) = E\lambda$. By letting $h = 1/x$, using the definition of $v_\kappa(x)$ in (3) and the definition of $\Psi$, one easily sees that $\Psi(h, \lambda) = (x, \kappa)$, and hence, $(x, \kappa)$ is in the image of $\Psi$, concluding the proof.

**Lemma 8**. *Let $(\mathcal{S}, \mathcal{R})$ be a reaction network such that the last two reactions $R_{r-1}$ and $R_r$ are reverse to each other. Let $(\tilde{\mathcal{S}}, \tilde{\mathcal{R}})$ be the reduced network obtained by removing the reaction $R_r$.*

*(a) The vector $(0, \ldots, 0, 1, 1) \in \mathbb{R}^r_{\geq 0}$ is an extreme vector of the flux cone of $(\mathcal{S}, \mathcal{R})$.*

*(b) If v is an extreme vector of the flux cone of $(\tilde{\mathcal{S}}, \tilde{\mathcal{R}})$, then $(v, 0)$ is an extreme vector of the flux cone of $(\mathcal{S}, \mathcal{R})$.*

*In particular, the flux cone of $(\mathcal{S}, \mathcal{R})$ has more extreme vectors than the flux cone of $(\tilde{\mathcal{S}}, \tilde{\mathcal{R}})$.*

*Proof.* Before starting the proof, we recall some definitions and statements. The support of a vector $u \in \mathbb{R}^r$ is the set of indices where the vector is non-zero: $\text{supp}(u) \coloneqq \{i \in \{1, \ldots, r\} \mid u_i \neq 0\}$. A vector $u \in \ker(N) \cap \mathbb{R}^r_{\geq 0}$ is said to have minimal support if for all $w \in \ker(N) \cap \mathbb{R}^r_{\geq 0}$ with $\text{supp}(w) \subseteq \text{supp}(u)$ we have $\text{supp}(w) = \text{supp}(u)$. A vector $u \in \ker(N) \cap \mathbb{R}^r_{\geq 0}$ is an extreme vector if and only if it has minimal support [70, Proposition 5], see also [71, Definition 1] and [72, Proposition 5].

(a) Since the reactions $R_{r-1}$ and $R_r$ are reverse to each other, for the last two columns of $N$ it holds that $N_{r-1} = -N_r$, which implies

$$N(0, \ldots, 0, 1, 1)^\top = N_{r-1} + N_r = 0.$$

Thus, $(0, \ldots, 0, 1, 1)$ belongs to the flux cone. If $(0, \ldots, 0, 1, 1)$ did not have minimal support, then $(0, \ldots, 0, 1)$ or $(0, \ldots, 0, 1, 0)$ would be contained in the flux cone, but that would imply that $N_r = 0$ and $N$ has a zero column, which cannot be the case.

(b) Let $\tilde{N} \in \mathbb{R}^{n \times (r-1)}$ denote the stoichiometric matrix of the reduced network and hence $N = (\tilde{N} \, N_r) \in \mathbb{R}^{n \times r}$. Since

$$N \begin{pmatrix} v \\ 0 \end{pmatrix} = \tilde{N} v = 0,$$

we have that $(v, 0)$ is contained in the flux cone of $(\mathcal{S}, \mathcal{R})$. To show that $(v, 0)$ is an extreme vector, we show that it has minimal support. Let $w \in \ker(N) \cap \mathbb{R}^r_{\geq 0}$ such that $\mathrm{supp}(w) \subseteq \mathrm{supp}$ $((v, 0))$. Then $w_n = 0$, and hence $(w_1 \ldots, w_{r-1}) \in \ker(\tilde{N}) \cap \mathbb{R}^{r-1}_{\geq 0}$. Since $v$ is an extreme vector of $\ker(\tilde{N}) \cap \mathbb{R}^{r-1}_{\geq 0}$ and $\mathrm{supp}((w_1 \ldots, w_{r-1})) \subseteq \mathrm{supp}(v)$, it follows that $\mathrm{supp}(w) = \mathrm{supp}((w_1 \ldots, w_{r-1})) = \mathrm{supp}(v) = \mathrm{supp}((v, 0))$. Hence $(v, 0)$ has minimal support and is therefore an extreme vector.

## Proof of Theorem 2

Theorem 2 is a direct consequence of two technical lemmas. To state the lemmas, we use the notation of the main text, and additionally we write $\Omega \subseteq \mathbb{R}^r_{>0} \times \mathbb{R}^d$ for the parameter region of multistationarity:

$$\Omega := \{(\kappa, c) \in \mathbb{R}^r_{>0} \times \mathbb{R}^d \mid \# \, (V_\kappa \cap \mathcal{P}_c \cap \mathbb{R}^n_{>0}) \geq 2\}.$$

We will write $\Omega$ as the image of a subset of $\mathcal{V}$ under the map

$$\pi : \mathcal{V} \to \mathbb{R}^r_{>0} \times \mathbb{R}^d, \quad (x, \kappa) \mapsto (\kappa, Wx), \tag{19}$$

and then compare path-connected components. Recall that $W$ is a matrix of conservation relations.

By Theorem 1, $\Omega$ is closely related to the preimage of the negative real half-line under the critical function given in (8) for a parametrization $\Phi : \mathcal{D} \to \mathcal{V}$:

$$g \circ \Phi : \mathcal{D} \to \mathbb{R}.$$

So we introduce the set:

$$\Theta := g^{-1}(\mathbb{R}_{\leq 0}) \cap \pi^{-1}(\Omega) \subseteq \mathcal{V}. \tag{20}$$

We summarize all the relevant functions that play a role in the proof of Theorem 2 in the following diagram:

$$
\begin{array}{ccccccc}
(g \circ \Phi)^{-1}(\mathbb{R}_{<0}) & \hookrightarrow & \Phi^{-1}(\Theta) & \hookrightarrow & (g \circ \Phi)^{-1}(\mathbb{R}_{\leq 0}) & \hookrightarrow & \mathcal{D} \\
\downarrow{\scriptstyle \Phi} & & \downarrow{\scriptstyle \Phi} & & \downarrow{\scriptstyle \Phi} & & \downarrow{\scriptstyle \Phi} \quad \searrow^{g \circ \Phi} \\
g^{-1}(\mathbb{R}_{<0}) & \hookrightarrow & \Theta & \hookrightarrow & g^{-1}(\mathbb{R}_{\leq 0}) & \hookrightarrow & \mathcal{V} \xrightarrow{g} \mathbb{R} \\
& & \downarrow{\scriptstyle \pi} & & & & \downarrow{\scriptstyle \pi} \\
& & \Omega & & \hookrightarrow & & \mathbb{R}^r_{>0} \times \mathbb{R}^d.
\end{array}
$$

We denote by $b_0(X)$ the number of path-connected components of a set $X \subseteq \mathbb{R}^k$ [73, Definition 3.3.7]. Note that $X$ is path connected if and only if $b_0(X) = 1$.

**Lemma 9**. *For a dissipative reaction network without relevant boundary steady states, it holds that*

$$\pi(\Theta) = \Omega.$$

*In particular, $b_0(\Omega) \leq b_0(\Theta)$.*

*Proof.* By definition of $\Theta$, we have $\pi(\Theta) \subseteq \Omega$. To show the reverse inclusion, consider $(\kappa, c) \in \Omega$. All we need is to find a point $(x^*, \kappa)$ such that $\pi(x^*, \kappa) = (\kappa, c)$ and $g(x^*, \kappa) \leq 0$. From the definition of $V_\kappa$ in (5) and of $\mathcal{P}_c$ in (4), it follows that

$$\pi^{-1}(\kappa, c) = \{(x^*, \kappa) \in \mathcal{V} \mid x^* \in V_\kappa \cap \mathcal{P}_c \cap \mathbb{R}^n_{>0}\}.$$

Since $(\kappa, c)$ enables multistationarity as it belongs to $\Omega$, we have $\pi^{-1}(\kappa, c)$ has at least two elements and hence Theorem 1(A) cannot hold. It follows that $\pi^{-1}(\kappa, c)$ contains at least one point where $(x^*, \kappa)$ with $g(x^*, \kappa) \leq 0$. As by construction $\pi(x^*, \kappa) = (\kappa, c)$, we have the inclusion.

The second part follows from the fact that continuous images of path connected sets are path connected [73, Theorem 3.3.5].

**Lemma 10**. *Consider a dissipative reaction network without relevant boundary steady states. If the closure of $(g \circ \Phi)^{-1}(\mathbb{R}_{<0})$ equals $(g \circ \Phi)^{-1}(\mathbb{R}_{\leq 0})$, then*

$$b_0(\Theta) \leq b_0((g \circ \Phi)^{-1}(\mathbb{R}_{<0})).$$

*Proof.* For all $(x, \kappa) \in \mathcal{V}$ with $g(x, \kappa) < 0$, Theorem 1(B) gives that $\pi(x, \kappa) \in \Omega$. Thus, by definition of $\Theta$, it holds that

$$g^{-1}(\mathbb{R}_{<0}) \subseteq \Theta \subseteq g^{-1}(\mathbb{R}_{\leq 0}).$$

By taking preimages under $\Phi$, it follows that

$$(g \circ \Phi)^{-1}(\mathbb{R}_{<0}) \subseteq \Phi^{-1}(\Theta) \subseteq (g \circ \Phi)^{-1}(\mathbb{R}_{\leq 0}).$$

Since $(g \circ \Phi)^{-1}(\mathbb{R}_{\leq 0})$ is the closure of $(g \circ \Phi)^{-1}(\mathbb{R}_{<0})$, every point $\eta \in \Phi^{-1}(\Theta)$ is contained in the closure of $(g \circ \Phi)^{-1}(\mathbb{R}_{<0})$. Using the Curve Selecting Lemma [74], there exists a continuous path

$$\gamma : [0, 1] \to \mathbb{R}^n$$

such that $\gamma(0) = \eta$ and $\gamma((0, 1)) \subseteq (g \circ \Phi)^{-1}(\mathbb{R}_{<0})$. Thus, there exists a continuous path between $\eta$ and one of the path-connected components of $(g \circ \Phi)^{-1}(\mathbb{R}_{<0})$. Therefore, $b_0(\Phi^{-1}(\Theta)) \leq b_0((g \circ \Phi)^{-1}(\mathbb{R}_{<0}))$.

Since $\Phi$ is surjective, it follows that $\Phi(\Phi^{-1}(\Theta)) = \Theta$. Since a continuous image of a path connected set is path connected [73, Theorem 3.3.5], we conclude that $b_0(\Theta) \leq b_0(\Phi^{-1}(\Theta))$.

## Proof of Theorem 4

To prove Theorem 4, we need to show that under the hypotheses of the theorem, we have

$$b_0(\Omega) \leq b_0((\tilde{g} \circ \tilde{\Phi})^{-1}(\mathbb{R}_{<0})).$$

The proof is based on inductive application of the following statement.

**Proposition 11**. *Let $(\mathcal{S}, \mathcal{R})$ be a conservative reaction network without relevant boundary steady states. Assume that the species $X_n$ participates in exactly 3 reactions of the form*

$$a_1 X_1 + \cdots + a_{n-1} X_{n-1} \underset{\kappa_r}{\overset{\kappa_{r-1}}{\rightleftharpoons}} X_n \xrightarrow{\kappa_{r-2}} b_1 X_1 + \cdots + b_{n-1} X_{n-1}.$$

*Let $(\tilde{\mathcal{S}}, \tilde{\mathcal{R}})$ denote the reduced network obtained by removing the reaction corresponding to $\kappa_r$ and $\tilde{\Theta}$ denote the set in* (20) *for* $(\tilde{\mathcal{S}}, \tilde{\mathcal{R}})$. *It holds that*

$$b_0(\Theta) \leq b_0(\tilde{\Theta}).$$

*Proof.* For every object corresponding to a network, we write ˜ to indicate that it corresponds to the reduced network, e.g. the reaction rate constants in the reduced network are denoted by $\tilde{\kappa}_1, \ldots, \tilde{\kappa}_{r-1}$.

As in the proof of Lemma 8, the stoichiometric matrices $N$ and $\tilde{N}$ satisfy $N_{r-1} = -N_r$, $\tilde{N}_j = N_j$ for $j = 1, \ldots, r-1$, and $\mathrm{rk}(N) = \mathrm{rk}(\tilde{N})$. Recall that $W \in \mathbb{R}^{d \times n}$ is a row reduced full rank matrix such that $WN = 0$. It follows that $W\tilde{N} = 0$, so $W$ is a matrix of conservation relations for $(\tilde{\mathcal{S}}, \tilde{\mathcal{R}})$ and we use this matrix in the definition of $\tilde{\pi}$, analogously to (19). Using (9), we also have that $(\tilde{\mathcal{S}}, \tilde{\mathcal{R}})$ is conservative if and only if $(\mathcal{S}, \mathcal{R})$ is conservative.

Following the proof of [75, Prop. 1] we introduce the maps:

$$\eta : \mathbb{R}^r_{>0} \to \mathbb{R}^{r-1}_{>0}, \quad \kappa \mapsto \left(\kappa_1, \ldots, \kappa_{r-2}, \frac{\kappa_{r-1}}{\kappa_r + \kappa_{r-2}}\right),$$

$$\tilde{\eta} : \mathbb{R}^{r-1}_{>0} \to \mathbb{R}^{r-1}_{>0}, \quad \tilde{\kappa} \mapsto \left(\kappa_1, \ldots, \kappa_{r-2}, \frac{\tilde{\kappa}_{r-1}}{\tilde{\kappa}_{r-2}}\right).$$

For $x \in \mathbb{R}^n$, let $x^a = x_1^{a_1} \cdots x_{n-1}^{a_{n-1}}$ where $a = (a_1, \ldots, a_{n-1}, 0)$.

We start by relating the steady states of the two networks under the hypothesis that $\eta(\kappa) = \tilde{\eta}(\tilde{\kappa})$. Recall that $(\kappa, c) \in \Omega$ if and only if the equation system

$$f_\kappa(x) = Nv_\kappa(x) = 0, \quad Wx = c \tag{21}$$

has at least two positive solutions. Redundant linear relations among the equations $f_\kappa(x) = 0$ arise from linear dependencies of the rows of $N$. To remove these redundancies, we consider $I = \{i_1, \ldots, i_d\}$ to be the set of indices of the first non-zero coordinates of each row of $W$, $i_1 < \cdots < i_d$. For all $j = 1, \ldots, d$, we replace the $i_j$th row of $f_\kappa(x)$ by the $j$th row of $Wx - c$. If we denote the resulting function by $h_{\kappa,c}(x)$, then $x^* \in \mathbb{R}^n_{>0}$ is a solution of (21) if and only if $h_{\kappa,c}(x^*) = 0$ holds. Thus, the parameter pair $(\kappa, c)$ enables multistationarity if and only if $h_{\kappa,c}(x) = 0$ has at least two positive solutions.

Since $x_n$ appears linearly in $h_{\kappa,c}(x)$, there exist vectors $z(\kappa)$, $v(x, \kappa)$ such that

$$h_{\kappa,c}(x) = \begin{pmatrix} z(\kappa) \\ -(\kappa_r + \kappa_{r-2}) \end{pmatrix} x_n + \begin{pmatrix} v(x, \kappa) \\ \kappa_{r-1} x^a \end{pmatrix}. \tag{22}$$

Specifically:

if $i = i_j \in I : z_i(\kappa) = W_{j,n}, \quad v_i(x, \kappa) = -c_j + W_{j,1}x_1 + \cdots + W_{j,n-1}x_{n-1}$,

if $i \in \{1, \ldots, n-1\} \setminus I : z_i(\kappa) = \kappa_r a_i + \kappa_{r-2}b_i, \quad v_i(x, \kappa) = -\kappa_{r-1}a_i x^a + u_i(x, \kappa)$,

where $u(x, \kappa)$ is chosen such that (22) holds.

We define $\tilde{h}_{\tilde{\kappa},c}(x)$ analogously for the reduced network and write

$$\tilde{h}_{\tilde{\kappa},c}(x) = \begin{pmatrix} \tilde{z}(\tilde{\kappa}) \\ -\tilde{\kappa}_{r-2} \end{pmatrix} x_n + \begin{pmatrix} \tilde{v}(x, \tilde{\kappa}) \\ -\tilde{\kappa}_{r-1}x^a \end{pmatrix},$$

which under the assumption that $\eta(\kappa) = \tilde{\eta}(\tilde{\kappa})$, we have

$$\text{if } i = i_j \in I: \quad \tilde{z}_i(\tilde{\kappa}) = z_i(\kappa), \quad \tilde{v}_i(x, \tilde{\kappa}) = v_i(x, \kappa),$$

$$\text{if } i \in \{1, \ldots, n-1\} \setminus I: \quad \tilde{z}_i(\tilde{\kappa}) = \tilde{\kappa}_{r-2} b_i, \quad \tilde{v}_i(x, \tilde{\kappa}) = -\tilde{\kappa}_{r-1} a_i x^a + u_i(x, \tilde{\kappa}).$$

It then holds that

$$\frac{\kappa_{r-1}}{\kappa_r + \kappa_{r-2}} z(\kappa) x^a + v(x, \kappa) = \frac{\tilde{\kappa}_{r-1}}{\tilde{\kappa}_{r-2}} \tilde{z}(\tilde{\kappa}) x^a + \tilde{v}(x, \tilde{\kappa}). \tag{23}$$

For $i \in I$, it is straightforward to see that this equality holds. If $i \notin I$, then the equation reduces to the equality

$$\frac{\kappa_{r-1}}{\kappa_r + \kappa_{r-2}} (\kappa_r a_i + \kappa_{r-2} b_i) - \kappa_{r-1} a_i = \frac{\kappa_{r-1}}{\kappa_r + \kappa_{r-2}} \kappa_{r-2} (b_i - a_i) = \frac{\tilde{\kappa}_{r-1}}{\tilde{\kappa}_{r-2}} \tilde{\kappa}_{r-2} (b_i - a_i)$$

$$= \tilde{\kappa}_{r-1} b_i - \tilde{\kappa}_{r-1} a_i.$$

With this in place, consider the matrices

$$B(\kappa) = \begin{pmatrix} \text{Id}_{n-1} & \frac{z(\kappa)}{\kappa_r + \kappa_{r-2}} \\ 0 & -\frac{1}{\kappa_r + \kappa_{r-2}} \end{pmatrix}, \text{ and } \tilde{B}(\tilde{\kappa}) = \begin{pmatrix} \text{Id}_{n-1} & \frac{\tilde{z}(\tilde{\kappa})}{\tilde{\kappa}_{r-2}} \\ 0 & -\frac{1}{\tilde{\kappa}_{r-2}} \end{pmatrix},$$

where $\text{Id}_{n-1}$ is the identity matrix of size $n-1$. Using (23), we have

$$B(\kappa) h_{\kappa,c}(x) = \begin{pmatrix} \text{Id}_{n-1} & \frac{z(\kappa)}{\kappa_r + \kappa_{r-2}} \\ 0 & -\frac{1}{\kappa_r + \kappa_{r-2}} \end{pmatrix} \begin{pmatrix} z(\kappa) x_n + v(x, \kappa) \\ -(\kappa_r + \kappa_{r-2}) x_n + \kappa_{r-1} x^a \end{pmatrix} =$$

$$\begin{pmatrix} \frac{\kappa_{r-1}}{\kappa_r + \kappa_{r-2}} z(\kappa) x^a + v(x, \kappa) \\ x_n - \frac{\kappa_{r-1}}{\kappa_r + \kappa_{r-2}} x^a \end{pmatrix} = \begin{pmatrix} \frac{\tilde{\kappa}_{r-1}}{\tilde{\kappa}_{r-2}} \tilde{z}(\tilde{\kappa}) x^a + \tilde{v}(x, \tilde{\kappa}) \\ x_n - \frac{\tilde{\kappa}_{r-1}}{\tilde{\kappa}_{r-2}} x^a \end{pmatrix} = \tilde{B}(\tilde{\kappa}) \tilde{h}_{\tilde{\kappa},c}(x). \tag{24}$$

Since the matrices $B(\kappa)$ and $\tilde{B}(\tilde{\kappa})$ are invertible, it follows from (24) that every positive solution of $h_{\kappa,c}(x) = 0$ is a solution of $h_{\tilde{\kappa},c}(x) = 0$. In particular, the reduced network does not have relevant boundary steady states.

Additionally, since $s = \tilde{s}$, and the Jacobian of $h_{\kappa,c}(x)$ is precisely the matrix $M_\kappa(x)$ (and analogous for the reduced network), taking the Jacobian and determinant of both sides of (24) yields:

$$-\frac{1}{\kappa_r + \kappa_{r-2}} g(x, \kappa) = -\frac{1}{\tilde{\kappa}_{r-2}} \tilde{g}(x, \tilde{\kappa}). \tag{25}$$

Since $\kappa_r, \kappa_{r-2}, \tilde{\kappa}_{r-2} > 0$, it follows that $g(x, \kappa)$ and $\tilde{g}(x, \tilde{\kappa})$ have the same sign if $\eta(\kappa) = \tilde{\eta}(\tilde{\kappa})$.

Finally, we can easily see that for $(x^*, \kappa) \in \mathcal{V}$ and $(x^*, \tilde{\kappa}) \in \tilde{\mathcal{V}}$ with $\eta(\kappa) = \tilde{\eta}(\tilde{\kappa})$ the following holds:

$$\pi(x^*, \kappa) \in \Omega \text{ if and only if } \tilde{\pi}(x^*, \tilde{\kappa}) \in \tilde{\Omega}. \tag{26}$$

For the forward implication, if $\pi(x^*, \kappa) \in \Omega$, then by definition, the parameter pair $(\kappa, c)$ with $c = W x^*$ enables multistationarity, that is $h_{\kappa,c}(x) = 0$ has at least two positive solutions. Hence so does $h_{\tilde{\kappa},c}(x) = 0$, and $(\tilde{\kappa}, c) = \tilde{\pi}(x^*, \tilde{\kappa}) \in \tilde{\Omega}$. The reverse implication of (26) follows analogously.

With these preliminaries in place, consider the maps

$$F: \quad \mathcal{V} \rightarrow \mathbb{R}_{>0}^{n-1} \times \mathbb{R}_{>0}^{r-1}, \quad (x, \kappa) \mapsto ((x_1, \cdots, x_{n-1}), \eta(\kappa)),$$
$$\tilde{F}: \quad \tilde{\mathcal{V}} \rightarrow \mathbb{R}_{>0}^{n-1} \times \mathbb{R}_{>0}^{r-1}, \quad (\tilde{x}, \tilde{\kappa}) \mapsto ((\tilde{x}_1, \cdots, \tilde{x}_{n-1}), \tilde{\eta}(\tilde{\kappa})).$$

By considering the steady state equation of $X_n$, for $(x, \kappa) \in \mathcal{V}$ and $(\tilde{x}, \tilde{\kappa}) \in \tilde{\mathcal{V}}$ the $n$th coordinate is uniquely determined by the rest of the coordinates, that is:

$$x_n = \frac{\kappa_{r-1}}{\kappa_r + \kappa_{r-2}} x^a, \quad \tilde{x}_n = \frac{\tilde{\kappa}_{r-1}}{\tilde{\kappa}_{r-2}} \tilde{x}^a. \tag{27}$$

This implies that

$$F(x, \kappa) = \tilde{F}(\tilde{x}, \tilde{\kappa}) \Leftrightarrow \begin{cases} x = \tilde{x} \\ \eta(\kappa) = \tilde{\eta}(\tilde{\kappa}). \end{cases} \tag{28}$$

Additionally, the restriction of $F$ to a fixed $\kappa$ or of $\tilde{F}$ to a fixed $\tilde{\kappa}$ are injective functions.

Now, the first step towards the proof of the theorem is to show that $\Theta$ satisfies

$$\Theta = F^{-1}(\tilde{F}(\tilde{\Theta})). \tag{29}$$

To show the inclusion $\subseteq$, let $(x, \kappa) \in \Theta \subseteq \mathcal{V}$. By the definition of $\Theta$ in (20), it holds $\pi(x, \kappa) \in \Omega$ and $g(x, \kappa) \leq 0$. Since $\tilde{\eta}$ is surjective, there exists $\tilde{\kappa} \in \mathbb{R}_{>0}^{r-1}$ such that $\eta(\kappa) = \tilde{\eta}(\tilde{\kappa})$. As $(x, \kappa) \in \mathcal{V}$, we have $(x, \tilde{\kappa}) \in \tilde{\mathcal{V}}$ by (24), and hence $\tilde{\pi}(x, \tilde{\kappa}) \in \tilde{\Omega}$ by (26). Now (25) implies also that $\tilde{g}(x, \tilde{\kappa}) \leq 0$, thus $(x, \tilde{\kappa}) \in \tilde{\Theta}$ by definition. As $F(x, \kappa) = \tilde{F}(x, \tilde{\kappa})$, the inclusion $\subseteq$ follows.

For the reverse inclusion $\supseteq$, let $(x, \kappa) \in F^{-1}(\tilde{F}(\tilde{\Theta}))$. Then there exists $(\tilde{x}, \tilde{\kappa}) \in \tilde{\Theta}$ such that $F(x, \kappa) = \tilde{F}(\tilde{x}, \tilde{\kappa})$. By (28) it follows that $x = \tilde{x}$ and $\eta(\kappa) = \tilde{\eta}(\tilde{\kappa})$. We now use (26) and (25) again, to show that $(x, \kappa) \in \Theta$.

From (29) follows that $F(x, \kappa) \in \tilde{F}(\tilde{\Theta})$ for all $(x, \kappa) \in \Theta$. As a next step of the proof, we show that there exists a continuous path between any two points $(x, \kappa), (x', \kappa') \in \Theta$, if $F(x, \kappa)$ and $F(x', \kappa')$ lie in the same path-connected component of $\tilde{F}(\tilde{\Theta})$. So let

$$\gamma : [0, 1] \rightarrow \mathbb{R}_{>0}^{n-1} \times \mathbb{R}_{>0}^{r-1}, \quad t \mapsto (y(t), \alpha(t))$$

be a continuous path such that $\gamma(0) = F(x, \kappa)$, $\gamma(1) = F(x', \kappa')$ and im $\gamma \subset \tilde{F}(\tilde{\Theta})$. We now extend $\gamma$ to a path in $\mathbb{R}_{>0}^n \times \mathbb{R}_{>0}^r$ by defining

$$\Gamma : [0, 1] \quad \rightarrow \mathbb{R}_{>0}^n \times \mathbb{R}_{>0}^r,$$
$$t \quad \mapsto ((y(t), \alpha_{r-1}(t)y(t)^a), (\alpha_1(t), \cdots, \alpha_{r-2}(t), \alpha_{r-1}(t)(\kappa_r + \alpha_{r-2}(t)), \kappa_r)).$$

Here $\kappa_r$ is fixed, and is the last component of the original parameter vector $\kappa$. Using $\gamma(0) = F(x, \kappa)$, that is, $y(0) = (x_1, \ldots, x_{n-1})$ and $\alpha(0) = \eta(\kappa)$, and the recovery property of the $n$th coordinate (27), we have

$$\Gamma(0) = \left( \left( x_1, \ldots, x_{n-1}, \frac{\kappa_{r-1}}{\kappa_r + \kappa_{r-2}} x^a \right), \left( \kappa_1, \ldots, \kappa_{r-2}, \frac{\kappa_{r-1}}{\kappa_r + \kappa_{r-2}} (\kappa_r + \kappa_{r-2}), \kappa_r \right) \right) = (x, \kappa).$$

Furthermore, im $\Gamma \subseteq F^{-1}(\tilde{F}(\tilde{\Theta})) = \Theta$, since for all $t \in [0, 1]$:

$$F(\Gamma(t)) = \left( y(t), \left( \alpha_1(t), \ldots, \alpha_{r-2}(t), \frac{\alpha_{r-1}(t)(\kappa_r + \alpha_{r-2}(t))}{\kappa_r + \alpha_{r-2}(t)} \right) \right) = \gamma(t) \in \tilde{F}(\tilde{\Theta}). \tag{30}$$

We have now connected $\gamma(0)$ to the point $\Gamma(1)$ via a continuous path in $\Theta$. We now construct a continuous path between $\Gamma(1)$ and $(x', \kappa')$ in $\Theta$. First note that by (30),

$$F(\Gamma(1)) = \gamma(1) = F(x', \kappa').$$

Thus, both $\Gamma(1)$ and $(x', \kappa')$ are contained in the set $F^{-1}(\gamma(1))$, which is

$$\{(x', (\kappa'_1, \ldots, \kappa'_{r-2}, \alpha_{r-1}(1)(\beta + \kappa'_{r-2}), \beta)) \mid \beta \in \mathbb{R}_{>0}\}.$$

Since this set is path connected, there exists a continuous path

$$\Lambda : [0, 1] \to F^{-1}(\gamma(1))$$

such that $\Lambda(0) = \Gamma(1)$ and $\Lambda(1) = (x', \kappa')$. This path is contained in $\Theta = F^{-1}(\tilde{F}(\tilde{\Theta}))$, since $F(\Lambda(t)) = \gamma(1) \in \tilde{F}(\tilde{\Theta})$ for all $t \in [0, 1]$.

To finish the proof of the proposition, let $U_1, \ldots, U_k$ be the path-connected components of $\Theta$ and let $\tilde{U}_1, \ldots, \tilde{U}_{\tilde{k}}$ be the path-connected components of $\tilde{\Theta}$. Since a continuous image of a path connected set is path connected, $\tilde{F}(\tilde{U}_1), \ldots, \tilde{F}(\tilde{U}_{\tilde{k}})$ are path connected. Moreover, from $\tilde{\Theta} = \cup_j \tilde{U}_j$ follows that $\tilde{F}(\tilde{\Theta}) = \cup_j \tilde{F}(\tilde{U}_j)$.

If $k > \tilde{k}$, then there must exist $j \in \{1, \ldots, \tilde{k}\}$ and $i_1 \neq i_2 \in \{1, \ldots, k\}$ and $(x, \kappa) \in U_{i_1}, (x', \kappa') \in U_{i_2}$ such that $F(x, \kappa), F(x', \kappa') \in \tilde{F}(\tilde{U}_j)$. By the above argument, there exist a continuous path between $(x, \kappa)$ and $(x', \kappa')$. This contradicts that $i_1 \neq i_2$. Therefore, $b_0(\Theta) = k \leq \tilde{k} = b_0(\tilde{\Theta})$ as desired.

*Proof of Theorem 4.* We conclude the proof of Theorem 4 using Proposition 11. For $i = 1, \ldots, k$, let $(\mathcal{S}_i, \mathcal{C}_i)$ denote the reaction network obtained by removing the reverse reactions corresponding to $j = 1, \ldots, i$. Furthermore, let $\tilde{\Theta}_i$ be the set corresponding to the network $(\mathcal{S}_i, \mathcal{C}_i)$ as defined in (20).

Since the closure of $(\tilde{g} \circ \tilde{\Phi})^{-1}(\mathbb{R}_{<0})$ equals $(\tilde{g} \circ \tilde{\Phi})^{-1}(\mathbb{R}_{\leq 0})$,

$$b_0(\tilde{\Theta}_k) \leq b_0((\tilde{g} \circ \tilde{\Phi})^{-1}(\mathbb{R}_{<0}))$$

by Lemma 10. Applying Proposition 11 inductively, one has that

$$b_0(\Theta) \leq b_0(\tilde{\Theta}_1) \leq \cdots \leq b_0(\tilde{\Theta}_k).$$

Now, we use Lemma 9 to conclude that the multistationarity region $\Omega$ of the network $(\mathcal{S}, \mathcal{R})$ satisfies:

$$b_0(\Omega) \leq b_0((\tilde{g} \circ \tilde{\Phi})^{-1}(\mathbb{R}_{<0})).$$

## Number of path-connected components of the multistationarity region for Fig 2(c)

We aim at showing that the multistationarity region $\Omega$ of the network in Fig 2(c) has exactly two path-connected components. We do this in two steps. First, we show that $b_0(\Omega) \geq 2$, and then that $b_0(\Omega) \leq 2$, giving the equality.

**Showing that $b_0(\Omega) \geq 2$.** We choose the order of species S, Sp, K, P, KL, PL, SKL, SpP, L, giving the following stoichiometric matrix, and choice of matrix of conservation relations:

$$N = \begin{bmatrix} -1 & 1 & 0 & 0 & 0 & 1 & 0 & 0 & 0 & 0 \\ 0 & 0 & 1 & -1 & 1 & 0 & 0 & 0 & 0 & 0 \\ 0 & 0 & 0 & 0 & 0 & 0 & -1 & 1 & 0 & 0 \\ 0 & 0 & 0 & -1 & 1 & 1 & 0 & 0 & -1 & 1 \\ -1 & 1 & 1 & 0 & 0 & 0 & 1 & -1 & 0 & 0 \\ 0 & 0 & 0 & 0 & 0 & 0 & 0 & 0 & 1 & -1 \\ 1 & -1 & -1 & 0 & 0 & 0 & 0 & 0 & 0 & 0 \\ 0 & 0 & 0 & 1 & -1 & -1 & 0 & 0 & 0 & 0 \\ 0 & 0 & 0 & 0 & 0 & 0 & -1 & 1 & -1 & 1 \end{bmatrix},$$

$$W = \begin{bmatrix} 1 & 1 & 0 & 0 & 0 & 0 & 1 & 1 & 0 \\ 0 & 0 & 1 & 0 & 1 & 0 & 1 & 0 & 0 \\ 0 & 0 & 0 & 1 & 0 & 1 & 0 & 1 & 0 \\ 0 & 0 & 0 & 0 & 1 & 1 & 1 & 0 & 1 \end{bmatrix}.$$

The network is conservative, consistent, and has no relevant boundary steady states (determined using the siphon criterion). Solving the steady states equations for $x_2, x_6, x_7, x_8, x_9$ gives that any positive steady state satisfies

$$x_2 = \frac{(\kappa_5 + \kappa_6)\kappa_1 \kappa_3 x_1 x_5}{\kappa_6(\kappa_2 + \kappa_3)\kappa_4 x_4}, \quad x_6 = \frac{\kappa_8 \kappa_9 x_4 x_5}{\kappa_7 \kappa_{10} x_3}, \quad x_7 = \frac{\kappa_1 x_1 x_5}{\kappa_2 + \kappa_3}, \quad x_8 = \frac{\kappa_1 \kappa_3 x_1 x_5}{\kappa_6(\kappa_2 + \kappa_3)}, \quad x_9 = \frac{\kappa_8 x_5}{\kappa_7 x_3}.$$

It is convenient to introduce the Michaelis-Menten constants of the two enzymatic processes of the network:

$$K_1 = \frac{\kappa_2 + \kappa_3}{\kappa_1}, \quad K_2 = \frac{\kappa_5 + \kappa_6}{\kappa_4}.$$

With this notation, and by letting $\xi$ be the vector of free variables, that is $\xi_1 = x_1, \xi_2 = x_3, \xi_3 = x_4, \xi_4 = x_5$, we obtain the parametrization $\Phi : \mathbb{R}^4_{>0} \times \mathbb{R}^{10}_{>0} \to \mathcal{V}$ given by

$$\Phi(\xi, \kappa) = \left( \xi_1, \frac{\kappa_3 K_2 \xi_1 \xi_4}{\kappa_6 K_1 \xi_3}, \xi_2, \xi_3, \xi_4, \frac{\kappa_8 \kappa_9 \xi_3 \xi_4}{\kappa_7 \kappa_{10} \xi_2}, \frac{\xi_1 \xi_4}{K_1}, \frac{\kappa_3 \xi_1 \xi_4}{\kappa_6 K_1}, \frac{\kappa_8 \xi_4}{\kappa_7 \xi_2} \right).$$

The critical function $g \circ \Phi$ is a quotient of polynomials with denominator $K_1 \kappa_6 \kappa_7 \xi_2^2 \xi_3$, and numerator a multiple of $\kappa_1 \kappa_4$. Since these expressions are positive for all positive $\xi$ and $\kappa$, the sign of $(g \circ \Phi)(\xi, \kappa)$ depends only on the sign of the numerator divided by $\kappa_1 \kappa_4$, which we denote by $p_\kappa(\xi)$. If we view $p_\kappa(\xi)$ as a polynomial in $\xi_1, \xi_2, \xi_3, \xi_4$, then there are two monomials $\xi_1 \xi_2^2 \xi_3^2 \xi_4$ and $\xi_1 \xi_2^2 \xi_3 \xi_4^2$ with coefficients

$$\kappa_6 \kappa_7 \kappa_8 \kappa_9 K_1 (\kappa_6 - \kappa_3), \text{ and } \kappa_3 \kappa_7 \kappa_8 \kappa_9 K_2 (\kappa_3 - \kappa_6).$$

The coefficient of the other monomials of $p_\kappa(\xi)$ are sums of products of the parameters, and

are positive. Thus, the value of the critical function $(g \circ \Phi)(\xi, \kappa)$ is positive for all $\xi \in \mathbb{R}_{>0}^4$ if $\kappa_3 = \kappa_6$. Now, one can apply part A of Theorem 1 to conclude that $(\kappa, c)$ does not enable multistationarity if $\kappa_3 = \kappa_6$.

As indicated in the main text, it is enough to show now that in the cases $\kappa_3 < \kappa_6$ and $\kappa_3 > \kappa_6$, the network can be multistationary. We show it is possible to choose $K_1, K_2, \kappa_1, \kappa_4, \kappa_7, \kappa_8,$ $\kappa_9, \kappa_{10}$ such that the polynomial $p_\kappa(\xi)$ takes negative values for some $\xi \in \mathbb{R}_{>0}^4$. To this end, we need to employ some standard techniques relating the signs a polynomial attains and its Newton polytope, and refer the reader for example to [48, Section 2.2]. Since the negative monomials of $p_\kappa(\xi)$ are contained in a face of the Newton polytope of $p_\kappa(\xi)$, it is enough to show that $p_\kappa(\xi)$ restricted to that face takes negative values. The restricted polynomial is given by $\kappa_1 \kappa_7 \xi_1 \xi_2$ times

$$q_\kappa(\xi) := \kappa_6 K_1 \xi_3^2 (\kappa_6 \kappa_7 \kappa_{10} \xi_2^2 + \kappa_8 \kappa_9 (\kappa_6 - \kappa_3) \xi_2 \xi_4 + \kappa_6 \kappa_8 \kappa_9 \xi_3 \xi_4)$$
$$+ \kappa_3 K_2 \xi_4^2 (\kappa_3 \kappa_7 \kappa_{10} \xi_2^2 + \kappa_8 \kappa_9 (\kappa_3 - \kappa_6) \xi_2 \xi_3 + \kappa_3 \kappa_8 \kappa_9 \xi_3 \xi_4).$$

This is a polynomial with exactly one negative coefficient, for any choice of $\kappa_3 \neq \kappa_6$.

If $\kappa_6 - \kappa_3 < 0$, by choosing $K_2$ very small (or $K_1$ large), the polynomial multiplying $K_1$ determines the sign of $q_\kappa(\xi)$. By letting $\xi_2 = 1, \xi_3 = \frac{1}{\xi_4}$, the polynomial multiplying $\kappa_6 K_1 \xi_3^2$ becomes

$$\kappa_6 \kappa_7 \kappa_{10} + \kappa_8 \kappa_9 (\kappa_6 - \kappa_3) \xi_4 + \kappa_6 \kappa_8 \kappa_9.$$

Hence, for any $\xi_4 > 0$ large enough, this polynomial is negative, and so is $q_\kappa(\xi)$. Therefore, $p_\kappa(\xi)$ also attains negative values.

If $\kappa_3 - \kappa_6 < 0$, all we need to do is to let $K_1$ be small enough, or $K_2$ large enough and repeat the argument. We conclude that $b_0(\Omega) \geq 2$.

**Showing that $b_0(\Omega) \leq 2$.** We now apply Theorem 4 to the reduced network:

$$S + KL \xrightarrow{\kappa_1} SKL \xrightarrow{\kappa_3} S_p + KL, \quad K + L \underset{\kappa_8}{\overset{\kappa_7}{\rightleftharpoons}} KL$$

$$S_p + P \xrightarrow{\kappa_2} S_p P \xrightarrow{\kappa_6} S + P, \quad P + L \underset{\kappa_{10}}{\overset{\kappa_9}{\rightleftharpoons}} PL.$$

A matrix of extreme vectors is

$$E = \begin{bmatrix} 0 & 0 & 1 \\ 0 & 0 & 1 \\ 0 & 0 & 1 \\ 0 & 0 & 1 \\ 0 & 1 & 0 \\ 0 & 1 & 0 \\ 1 & 0 & 0 \\ 1 & 0 & 0 \end{bmatrix}.$$

The critical polynomial $\tilde{g} \circ \tilde{\Phi}$ in the variables $(\lambda_1, \lambda_2, \lambda_3, h_1, \ldots, h_9)$ has 2 negative monomials corresponding to the exponents

$$\beta_1 := (1, 1, 3, 0, 1, 0, 1, 1, 0, 1, 0, 1), \ \beta_2 := (1, 1, 3, 1, 0, 0, 1, 1, 0, 0, 1, 1),$$

and it has 42 monomials with positive coefficients.

In the following, we write

$$(\tilde{g} \circ \tilde{\Phi})(y) = -c_{\beta_1} y^{\beta_1} - c_{\beta_2} y^{\beta_2} + \sum_{\alpha \in \sigma_+(\tilde{g} \circ \tilde{\Phi})} c_\alpha y^\alpha,$$

where $\sigma_+(\tilde{g} \circ \tilde{\Phi})$ denotes the set of positive monomials and $c_\alpha$ are all positive. For the vector $v = (-6, 0, 0, 0, 0, 0, 2, 2, 0, 0, 0, 2)$ it holds that

$$\begin{aligned} v \cdot \beta_1 = 0, \quad & v \cdot \beta_2 = 0, \\ v \cdot \alpha \le 0, \quad & \text{for all } \alpha \in \sigma_+(\tilde{g} \circ \tilde{\Phi}), \end{aligned} \tag{31}$$

where for exactly two monomials $\alpha_1, \alpha_2 \in \sigma_+(\tilde{g} \circ \tilde{\Phi})$ there is equality in (31). Define the polynomial

$$h(y) := c_{\alpha_1} y^{\alpha_1} + c_{\alpha_2} y^{\alpha_2} - c_{\beta_1} y^{\beta_1} - c_{\beta_2} y^{\beta_2}.$$

Using [18, Corollary 3.13], it follows that $b_0(h^{-1}(\mathbb{R}_{<0})) \le 2$.

The next step is to show that $b_0((\tilde{g} \circ \tilde{\Phi})^{-1}(\mathbb{R}_{<0})) \le b_0(h^{-1}(\mathbb{R}_{<0}))$. First, we observe that $(\tilde{g} \circ \tilde{\Phi})(y) \ge h(y)$ for all $y \in \mathbb{R}_{>0}^{12}$ and hence

$$(\tilde{g} \circ \tilde{\Phi})^{-1}(\mathbb{R}_{<0}) \subseteq h^{-1}(\mathbb{R}_{<0}). \tag{32}$$

If $y, y' \in (\tilde{g} \circ \tilde{\Phi})^{-1}(\mathbb{R}_{<0})$ belong to the same path-connected component of $h^{-1}(\mathbb{R}_{>0})$, then we build a path between $y$ and $y'$ contained in $(\tilde{g} \circ \tilde{\Phi})^{-1}(\mathbb{R}_{<0})$ as follows. Since $y$ and $y'$ lie in the same path-connected component of $h^{-1}(\mathbb{R}_{<0})$, by (32) there exists a continuous path

$$\gamma : [0, 1] \to h^{-1}(\mathbb{R}_{<0})$$

such that $\gamma(0) = y$, $\gamma(1) = y'$. Note that the image of $\gamma$ not necessarily contained in $(\tilde{g} \circ \tilde{\Phi})^{-1}(\mathbb{R}_{<0})$.

For each $s \in [0, 1]$, we define the function $f_s : \mathbb{R}_{>0} \to h^{-1}(\mathbb{R}_{<0})$ as

$$f_s(t) = (\tilde{g} \circ \tilde{\Phi})(\gamma(s) \circ t^v) = -c_{\beta_1} \gamma(s)^{\beta_1} t^{v \cdot \beta_1} - c_{\beta_2} \gamma(s)^{\beta_2} t^{v \cdot \beta_2} + \sum_{\alpha \in \sigma_+(\tilde{g} \circ \tilde{\Phi})} c_\alpha \gamma(s)^\alpha t^{v \cdot \alpha},$$

$$= h(\gamma(s)) + p_s(t),$$

where $\gamma(s) \circ t^v = (\gamma_1(s) t^{v_1}, \dots, \gamma_{12}(s) t^{v_{12}})$, and from (31) $p_s(t)$ is a sum of generalized monomials in $t$ with all exponents negative and all coefficients positive. Hence the leading coefficient of $f_s(t)$ is $h(\gamma(s)) < 0$, all the other coefficients of $f_s(t)$ are positive, and $f_s(t)$ has a unique positive real root $T_s > 0$. Since the roots of a polynomial depend continuously on the coefficients, $T_s$ depends continuously on $s$. Therefore, $T := \max_{s \in [0, 1]} T_s$ exists.

For each $s \in [0, 1]$ and $t_0 > \max\{1, T\}$ it holds that $(\tilde{g} \circ \tilde{\Phi})(\gamma(s) \circ t_0^v) = f_s(t_0) < 0$. Using this observation, we define the path:

$$\Gamma : [0, 1] \to (\tilde{g} \circ \tilde{\Phi})^{-1}(\mathbb{R}_{<0}), \quad s \mapsto \gamma(s) \circ t_0^v.$$

Now, we connect the points $y$, $\Gamma(0)$, and $y'$, $\Gamma(1)$ using respectively the paths:

$$\begin{aligned} \gamma_1 : [1, t_0] \to (\tilde{g} \circ \tilde{\Phi})^{-1}(\mathbb{R}_{<0}) \quad & \gamma_2 : [1, t_0] \to (\tilde{g} \circ \tilde{\Phi})^{-1}(\mathbb{R}_{<0}) \\ t \mapsto y \circ t^v \quad & t \mapsto y' \circ t^v. \end{aligned}$$

Indeed, $\gamma_1(1) = y \circ 1^v = y$, $\gamma_1(t_0) = y \circ t_0^v = \gamma(0) \circ t_0^v = \Gamma(0)$, and the image of $\gamma_1$ is contained in $(\tilde{g} \circ \tilde{\Phi})^{-1}(\mathbb{R}_{<0})$, as $(\tilde{g} \circ \tilde{\Phi})(y \circ t^v)$ is negative at $t = 1$, has negative leading coefficient, and has

at most one positive real root. The analogous argument shows that $\gamma_2$ connects $y'$ and $\Gamma(1)$ in $(\tilde{g} \circ \tilde{\Phi})^{-1}(\mathbb{R}_{<0})$.

The concatenated path $\gamma_2^{-1} \, \Gamma \, \gamma_1$ gives a continuous path from $y, y'$ contained in $(\tilde{g} \circ \tilde{\Phi})^{-1}(\mathbb{R}_{<0})$. We conclude that the number of path-connected components of $(\tilde{g} \circ \tilde{\Phi})^{-1}(\mathbb{R}_{<0})$ is less or equal than the number of path-connected components of $h^{-1}(\mathbb{R}_{<0})$. Hence the inequality $b_0((\tilde{g} \circ \tilde{\Phi})^{-1}(\mathbb{R}_{<0})) \leq b_0(h^{-1}(\mathbb{R}_{>0}))$ holds.

This worked for the reduced network. In order to lift the result to the original network, we need to verify the extra condition of Theorem 4, namely that the closure of $(\tilde{g} \circ \tilde{\Phi})^{-1}(\mathbb{R}_{<0})$ is $(\tilde{g} \circ \tilde{\Phi})^{-1}(\mathbb{R}_{\leq 0})$. To see this, it is enough to show that the gradient $\nabla(\tilde{g} \circ \tilde{\Phi})(y)$ is different from zero for all $y \in \mathbb{R}_{>0}^{12}$. Since the sixth entry of both $\beta_1$ and $\beta_2$ are zero and the exponent $(1, 1, 3, 1, 1, 1, 1, 0, 0, 1, 0, 0)$ corresponds to a positive monomial of $\tilde{g} \circ \tilde{\Phi}$, the last entry of $\nabla(\tilde{g} \circ \tilde{\Phi})(y)$ cannot be zero. Using Theorem 4 we conclude that:

$$b_0(\Omega) \leq b_0((\tilde{g} \circ \tilde{\Phi})^{-1}(\mathbb{R}_{<0})) \leq b_0(h^{-1}(\mathbb{R}_{<0})) \leq 2$$

as desired.

## Supporting information

**S1 Source Code. Code for the algorithm.** Jupyter notebook that contains the code of the algorithm, written in `SageMath` 9.2 [32].
(IPYNB)

## Acknowledgments

We thank Sebastian Manecke and Oskar Henriksson for useful discussions about extreme vectors, and Carsten Wiuf for comments on the manuscript.

## Author Contributions

**Conceptualization:** Máté László Telek, Elisenda Feliu.

**Formal analysis:** Máté László Telek, Elisenda Feliu.

**Investigation:** Máté László Telek, Elisenda Feliu.

**Software:** Máté László Telek.

**Supervision:** Elisenda Feliu.

**Writing – original draft:** Máté László Telek, Elisenda Feliu.

**Writing – review & editing:** Máté László Telek, Elisenda Feliu.

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
