## [Decision Letter · Decision Letter 0]

21 Dec 2022

Dear Dr Feliu,

Thank you very much for submitting your manuscript "Topological descriptors of the parameter region of multistationarity: deciding upon connectivity" for consideration at PLOS Computational Biology. As with all papers reviewed by the journal, your manuscript was reviewed by members of the editorial board and by several independent reviewers. The reviewers appreciated the attention to an important topic. Based on the reviews, we are likely to accept this manuscript for publication, providing that you modify the manuscript according to the review recommendations.

Sincerely,

Andrea Ciliberto

Academic Editor

PLOS Computational Biology

Lucy Houghton

Staff

PLOS Computational Biology

Reviewer's Responses to Questions

**Comments to the Authors:**

Reviewer #1: This work addresses the challenging problem for chemical reaction networks: deciding the connectivity of the parameter region of multistationarity.

In principle, the problem can be solved by a generic algorithm. However, the generic algorithms is not really practical for realistic networks since a chemical reaction network usually contains lots of parameters and variables. In fact, deciding the number of connected components for a semi-algebraic set is an open problem in computational algebraic geometry. This work provides an efficient and effective way for deciding the connectivity

of the multistationary region. Although their method is inconclusive sometimes, the computations show that for lots of important networks, their method is successful. Also, for the allosteric reciprocal enzyme regulation network, they prove that the multistationary region has exactly two connected components.

Below, we have some small comments/suggestions:

1. In the abstract and the introduction, it sounds like the number of the connected components for the allosteric reciprocal enzyme regulation network is also computed by the proposed algorithm based on linear programming. In fact, the algorithm only works for the networks that have a connected multistationariy region. So, the authors should

clarify this point in the introduction.

2. Maybe the authors can provide more details on how to carry out Step 3 in Algorithm 5 since this is an important step. Now, there is only one sentence "the code is given in the supporting information".

3. In the parametrization between the lines 965 and 966, the authors should explain what you mean by the notions $\\Xi_1, \\Xi_2, \\Xi_3,..$. Or, maybe you can simply use $x_1, x_3, x_4...$ (they are clear).

4. Since Theorem 4 is the main theorem of this work, you may want to show how important it is by the experimental results. I understand the discussion in Page 14. But you can still say more on how Theorem 4/Step 4 helps for the examples showing in Table 1.

Reviewer #2: The paper "Topological descriptors of the parameter region of multistationarity: deciding upon connectivity" addresses a challenging question arising modelling biological systems: for what values of the parameters (kinetic constants and total concentrations) is a model capable of exhibiting multistationarity. (Multistationarity itself is of interest in modeling the cell cycle and cellular decision making.) While there has recently been progress in developing algorithmic ways to establish multistationariy it is currently not possible to characterize the parameter region where this phenomenon occurs. In particular, there is currently no straightforward way to algorithmically decide upon the connectedness of this parameter region (at least for everybody that is not an expert in real algebraic geometry). This is unfortunate as connected components of the multistaionarity region have a distinct biological interpretation and hence offer biological insights that can only be obtained by mathematical modeling.

The present paper address this shortcoming and presents an algorithmic way of deciding connectedness that is accessible to a wide audience. It is very well written and easy to follow.

Two minor suggestions:

- in the formula between line 327 and 328 and in line 332, shouldn't it be f(R_{<0})?

- in line 997: maybe use c_\\alpha instead of c_*?

Reviewer #3: The manuscript under review investigates the region of multistationarity for multistationary mass-action systems in general. In particular, the authors aim at deciding whether this region is connected, and they cite the biological meaning of this.

The algorithm they propose is tested on a couple of well-known (usually enzymatic) reaction networks, and proves to be reasonably fast. The correctness of the algorithm is ensured by theorems that are proved in the appendix. Further, since in some cases the algorithm run for too long, the authors developed a tool that allows concluding on

the number of connected components by checking the same property for a certain reduced (and thus, smaller) network.

It is important to remark that the region of multistationarity is understood in the present work as a set with the rate constants and the conservation laws coupled. Another option would be to take only the rate constants (and ask if there exists a stoichiometric class with multiple steady states).

Another aspect important to keep in mind is that multistationarity does not imply multiple *stable* steady states. Multistability is certainly an important question, but it is beyond the scope of the current manuscript.

It turns out that for most biological networks examined by the authors, the region of multistationarity is connected (for only one network there is more than one connected component). It would be good to understand whether the exceptional network has some hidden motif that results in two connected components. In fact, I would see the point in finding further networks (possibly small toy examples) which show multiple connected components. This would go more in future work.

I find that the manuscript is well written, has a nice flow, and contains substantial results.

Minor typos:

Fig.1 caption line 2: "output IS the concentration"?

Fig.3 top right corner: "choices" vs. "choice"

line 38: 0th vs. 0-th

line 39: "Higher order Betti" vs. "Higher Betti"?

lines 101 and 127: "paper" vs. "manuscript"

line 109: "70s" vs. "70ies"

line 183: remove dash from "$i_j$-th" and "$j$-th"

line 207 (and elsewhere): "negative real half-line" vs. "negative real line"

line 232: maybe a word is missing from the sentence "We proceed to..."

line 233: "consists of" vs. "consists in"

line 240: "$i$th" vs. "$i$-th"

line 244: "all reactions producing them" vs. "reaction producing it"

line 259: why is "Methods" slanted?

formula after line 265: $i$ goes from $1$ to $\\ell$

line 277: is the term "consistent" standard? it is called "dynamically nontrivial" in "Banaji: Counting CRNs with NAUTY" and subsequent works

(to see the equivalence, apply the Farkas Lemma)

line 283: "coordinates" vs. "coordinate"

line 321: it sounds like the rational function might have polynomial denominator (solution: exchange the two sides of the "or", and adjust accordingly in the next sentence)

line 340: do you require that v is nonzero?

line 347: v=(2,3)?

line 426: not vs. no

line 466: "Since s=3, the negative of the ..."

line 470: (1,0,0,1,0,1) vs. (1,0,0,1,0,0)

line 535: "carried out by" vs. "carried by"?

line 549, 703: weakly vs. weak?

line 602: "Finally, we consider" vs. "We have finally consider"

line 686: Theorem 2 holds, because you prove it. Probably you wanted to phrase this sentence differently.

above line 887: an "=" is missing?

line 960: "no" vs. "not"

line 962: "state" vs. "states"

line 998: "=" vs. "=="

**Have the authors made all data and (if applicable) computational code underlying the findings in their manuscript fully available?**

Reviewer #1: None

Reviewer #2: Yes

Reviewer #3: Yes

PLOS authors have the option to publish the peer review history of their article (what does this mean?). If published, this will include your full peer review and any attached files.

Reviewer #1: No

Reviewer #2: No

Reviewer #3: No

Figure Files:

Data Requirements:

Reproducibility:

References:

---

## [Decision Letter · Decision Letter 1]

22 Feb 2023

Dear Dr Feliu,

We are pleased to inform you that your manuscript 'Topological descriptors of the parameter region of multistationarity: deciding upon connectivity' has been provisionally accepted for publication in PLOS Computational Biology. Please note the additional comments of Reviewer #3 below.

Best regards,

Andrea Ciliberto

Academic Editor

PLOS Computational Biology

Lucy Houghton

Staff

PLOS Computational Biology

Reviewer's Responses to Questions

**Comments to the Authors:**

Reviewer #1: The authors have answered all my questions. Also, they have revised the paper according to my comments.

Reviewer #3: Thank you, I am happy with the changes. The only typo I found is on line 735.

line 735: "On the other hand" vs. "On the other"

I disagree with the author's opinion on the non-equivalence of "consistent" and "dynamically nontrivial".

I think, these two notions are 100% equivalent. The "counterexample" given by the authors is wrong:

all solutions of a kinetic ODE with stoichiometric matrix N=(1,0) are monotonically increasing to infinity.

This is something that we want to call trivial (not nontrivial).

For a matrix N (by the Farkas' Lemma):

there is no strictly positive vector in the kernel of N

if and only if

the transpose of N has a vector in its image that is nonzero and nonnegative

(equivalently, there is a linear Lyapunov function).

And both of these are equivalent to dynamically trivial. Or equivalently, not consistent.

Anyway, this need not necessarily go the paper, as the paper deals with

an algebraic (or topological) question, and not the dynamics of kinetic ODEs.

**Have the authors made all data and (if applicable) computational code underlying the findings in their manuscript fully available?**

Reviewer #1: None

Reviewer #3: Yes

PLOS authors have the option to publish the peer review history of their article (what does this mean?). If published, this will include your full peer review and any attached files.

Reviewer #1: No

Reviewer #3: No

---

## [Editor Report · Acceptance letter]

19 Mar 2023

PCOMPBIOL-D-22-01491R1 

Topological descriptors of the parameter region of multistationarity: deciding upon connectivity

Dear Dr Feliu,

I am pleased to inform you that your manuscript has been formally accepted for publication in PLOS Computational Biology. Your manuscript is now with our production department and you will be notified of the publication date in due course.

With kind regards,

Zsofia Freund
